# FUSION-BASED ERROR-FEEDBACK AUGMENTATION FOR LOW-RESOURCE RADIOLOGY REPORT GENERATION

## ABSTRACT

Radiology report generation (RRG) is critical for assisting clinical diagnosis, yet current methods struggle to effectively integrate multi-view images and longitudinal patient data while operating under constrained annotation resources. Existing approaches often rely on large-scale supervised datasets and lack adaptability in low-resource settings. To address these challenges, we propose FEFA, a novel approach that combines a multi-expert image fusion module with an error-feedback augmentation strategy powered by large language models. Our fusion module dynamically combines current and prior images using tailored attention and gating mechanisms, producing compact and informative representations. Furthermore, the error-feedback mechanism enables self-correction during training by incorporating error analyses from previous stages. Experiments on MIMIC-CXR show that FEFA achieves state-of-the-art performance with only 8% of the supervised data, attaining 94% of the clinical efficacy score of the best existing method while outperforming all other competitors. Our work demonstrates significant improvements in data efficiency and model adaptability for real-world clinical scenarios.

## 1 INTRODUCTION

Automated radiology report generation (RRG) has gradually become a critical task in both clinical medicine and AI (Ke et al.; Sepehri et al., 2025; Xia et al., 2025). While most existing RRG methods focus solely on a single image when generating reports (Bu et al., 2024; Xiao et al., 2025), more and more works are trying to emulate the clinical workflow by leveraging multi-view information such as lateral images (Chen et al., 2020; Qin & Song, 2022), chief complaints (Nguyen et al., 2023; Liu et al., 2024), prior images (Serra et al., 2023; Wang et al., 2024) and corresponding reports (Hou et al., 2023a; Liu et al., 2025). However, there still remain several problems in this domain.

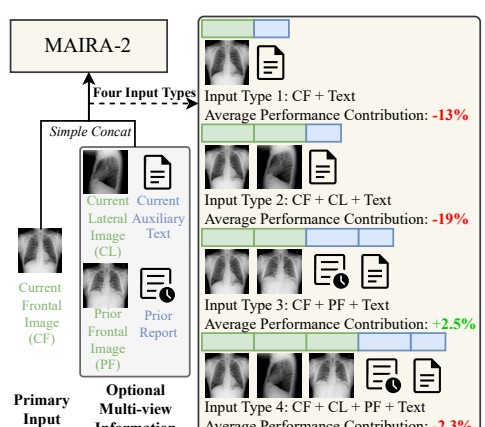

Figure 1: Four input types of MAIRA-2.

Due to concerns about patient privacy and the substantial human resources required for annotation, pairwise radiology report data are difficult to obtain. As a result, the supervised data available for RRG are inherently scarce, particularly for rare diseases. Although MIMIC-CXR (Johnson et al., 2019), the largest RRG dataset, has been widely used for training, this also leads existing methods to become overly reliant on large-scale supervised data, hindering their adaptability to more realistic low-resource scenarios. Additionally, training on the entire dataset is time-consuming and demands considerable memory and computing power. Therefore, we believe that we should mine deeper into the limited annotated medical images and report pairs, while leveraging information from various dimensions. Although the existing best approach MAIRA-2 (Bannur et al., 2024) proposes to integrate all available cues, its fusion method cannot effectively capture all important features of multi-view input images. For example, according to our pilot experiments, adding additional CL images (Input

Table 1: Results of our pilot experiment on MAIRA-2, which probes whether current lateral images (CL) can be exploited effectively. CX-14 refers to the classification performance across 14 clinical observations labeled by CheXbert and u→n denotes mapping the "uncertain" label to negative.

| Input Type | CX-14 (u→n) | | Input Type | CX-14 (u→n) | |
|---|---|---|---|---|---|
| | Mac-F1 | Mic-F1 | | Mac-F1 | Mic-F1 |
| CF+CL | 26.66 | 42.06 | CF+CL+PF | **39.81** | **51.71** |
| discard CL | **31.58** | **43.37** | discard CL | 38.83 | 50.30 |

Type 2 vs. Type 1 in Figure 1) actually leads to a decrease in performance when using the fusion schema of MAIRA-2.

To properly address above challenges, we propose a multi-expert image fusion module and introduce an error-feedback augmentation strategy based on Large Language Models (LLMs), namely FEFA. Concretely, we first design three distinct expert networks (see Figure 2) to process the images across four input types (see Figure 1). Current frontal or dual-view images are fused via self- or cross-attention mechanisms (Experts 1 & 2). When longitudinal data are available, a gated fusion is applied to the resulting current features and the prior frontal image (Expert 3). This yields a compact, fixed-length image feature sequence, rather than the variable-length concatenation produced by MAIRA-2. To enable robust training in low-resource scenarios, we adopt a two-stage curriculum learning strategy (see Figure 3). In stage 1, all prior reports are discarded, forcing the model to learn mainly from the current contents. In stage 2, available prior reports are reintroduced so that the model can exploit longitudinal cues. After that, our proposed error-feedback augmentation strategy is applied. Specifically, we compare the model predictions on the training set with ground truth to obtain sample-level error analyses. Each analysis is then incorporated as an additional textual input for a new stage-3 training. In our strategy, error analyses act as feedback from the previous training stage, prompting the model to correct potential mistakes during the current training round and thus perform implicit self-correction. At test time, we replace sample-level error analyses with four type-level common error lists, serving as a coarse-grained feedback prompt. Extensive experiments on MIMIC-CXR demonstrate the superiority of our FEFA with limited supervised data.

Our contributions are stated as follows: 1) We propose a multi-expert image fusion module that dynamically selects tailored feature fusion strategy based on the predefined input types. Compared to SOTA MAIRA-2, our approach yields features with a more uniform information density. 2) We introduce a novel error-feedback augmentation strategy that enables the model to implicitly learn self-correction during training on the target task, achieving additional performance gains. 3) Our FEFA is highly data-efficient: with only 8% of the supervised data used for SOTA MAIRA-2, it attains 94% of its clinical efficacy (CE) score and outperforms all other competitive baselines.

## 2 RELATED WORKS

**Radiology Report Generation (RRG).** The goal of RRG is to emulate the workflow of radiologists by interpreting radiological images and automatically producing detailed reports. In clinical practice, images from different views (frontal, lateral, etc.) along with patient-specific information (e.g., chief complaints, medical history) serve as crucial references for report writing. For example, lateral images are essential in identifying abnormalities of the hilum, as the mediastinum doesn't overlap hilar vessels in this view. However, most recent studies (Hou et al., 2023b; Bu et al., 2024; Xiao et al., 2025) still model RRG as a medical image captioning task, where a single frontal image is translated into textual descriptions. Despite their performance improvements, these approaches discard the remaining views and thus cannot provide comprehensive insights to patients.

To bridge this gap, a growing number of studies have attempted to incorporate information from different views. Among these, the lateral perspective was the earliest to be integrated. As a pioneering work, HRGR-Agent (Li et al., 2018) makes use of frontal and lateral images by averaging their features as the subsequent visual input. Later models either replicate this simple fusion (Chen et al., 2020; 2021; Qin & Song, 2022; Li et al., 2023b) or refine it with attention mechanisms (Yuan et al., 2019; Miura et al., 2021). Beyond the visual modality, Nguyen et al. (2023) and Liu et al. (2024) append the "INDICATION" field text to inject the patient's chief complaint. Alternatively, some studies focus on the use of longitudinal data, feeding prior images alone (Serra et al., 2023; Wang

et al., 2024) or with the report (Zhu et al., 2023; Hou et al., 2023a; Mei et al., 2024; Liu et al., 2025) to provide a comparative reference.

Unlike most works that incorporate only one extra view, MAIRA-2 (Bannur et al., 2024) is the first system to jointly ingest all available data (dual-view images, auxiliary texts, prior images with corresponding report) and sets a new state of the art. To preserve this advantage, we follow this setting, but design a new fusion strategy and adopt curriculum learning to handle the issues of its simple concatenation for all potential medical information. Moreover, we leverage a novel error-feedback augmentation to generate four lists of common errors at the type level. These lists support the model in understanding heterogeneous input medical features during the testing phase.

**Low-resource RRG.** A limited number of studies have addressed RRG tasks under low-resource conditions. SRFE (Li et al., 2024) exploits additional unpaired text to boost model performance when supervised data are scarce. By treating a large collection of unpaired textual reports as an external knowledge base, the approach retrieves reports that are semantically similar to the given image and utilizes the aligned multi-modal features for report generation. Meanwhile, RAMT (Zhang et al., 2024) makes full use of unpaired images by introducing the Mean Teacher paradigm to the RRG task for the first time. In this approach, a student network is trained with limited paired data, and forced to match the predictions of an EMA teacher when unpaired images come. Under this scheme, unpaired images are transformed into regularization signals that help stabilize the model. Compared to the aforementioned approaches, our model needs no extra unpaired data and yet achieves superior performance with less supervised data.

**Error Feedback in LLM.** Error feedback is commonly employed to guide large language models in achieving self-correction. Some works (Madaan et al., 2023; Shinn et al., 2023; Gou et al., 2024) acquire feedback only in the inference phase to refine the output. For instance, Self-Refine (Madaan et al., 2023) examines the output across multiple dimensions (including correctness, safety, fluency, etc.) to generate feedback, and subsequently optimizes the output accordingly. This feedback-refinement cycle can be repeated iteratively for several rounds. Alternatively, some research (Zelikman et al., 2022; An et al., 2023; Tong et al., 2024; Pan et al., 2025) utilizes error feedback for data augmentation to synthesize new training data. For example, LEMA (An et al., 2023) leverages GPT-4 to correct flawed reasoning paths generated by the original model, and retrains the model with the resulting self-correction demonstrations afterwards.

Unlike above studies, our error-feedback augmentation strategy treats the feedback as an extra input view for the RRG task. In this way, we can achieve implicit self-correction during the target task training, without requiring a more powerful teacher model or multi-step reasoning.

## 3 METHODOLOGY

### 3.1 PROBLEM FORMULATION

We mainly follow the SOTA MAIRA-2's pipeline. Let $\mathcal{D}_{tr} = \{(x_i^{fro}, x_i^{lat}, z_i, x_i^{pri}, y_i^{pri}, y_i)\}_{i=1}^n$ be the training set, where $n$ denotes the total number of samples. Each sample consists of a current frontal image $x_i^{fro}$, a current lateral image $x_i^{lat}$ (which may be absent), an auxiliary text $z_i$ such as "INDICATION" (which may be absent), a prior frontal image $x_i^{pri}$ with its report $y_i^{pri}$ (which may be absent), and a reference report $y_i$. Our goal is to learn the function $F_\theta(\cdot)$ that maps $(x_i^{fro}, x_i^{lat}, z_i, x_i^{pri}, y_i^{pri})$ to $y_i$ on the training set $\mathcal{D}_{tr}$, such that $F_\theta(x_i^{fro}, x_i^{lat}, z_i, x_i^{pri}, y_i^{pri}) \to y_i$, when $n$ is not too large.

### 3.2 BACKGROUND: MAIRA-2's FUSION FLAW

MAIRA-2 is a LLaVA-based model that fuses multi-view data via simple concatenation. Since not all samples contain complete multi-view data, the input can be categorized into four distinct types based on the combination of available image views (e.g., CF, CF+CL+PF, etc.) as shown in Figure 1. Although MAIRA-2 achieves impressive overall results, we observe that the performance between input types varies dramatically. It seems that samples including lateral views consistently underperform their frontal-only counterparts, regardless of whether longitudinal information is present (i.e., CF outperforms CF+CL, CF+PF outperforms CF+CL+PF). To rule out sample bias, we ran a

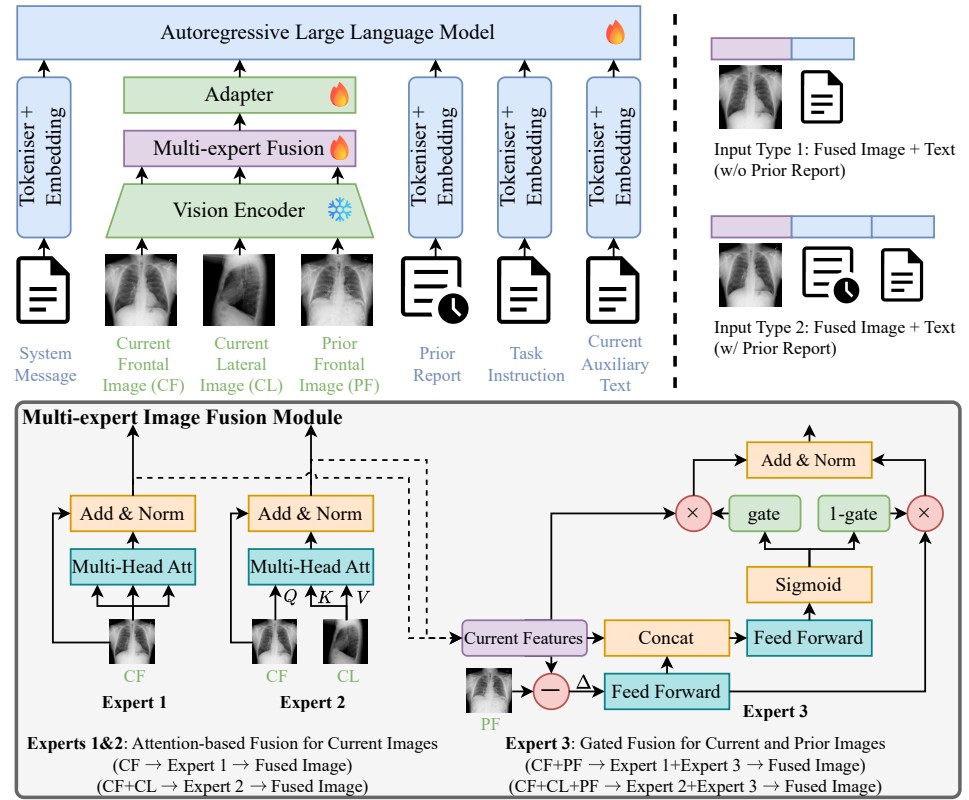

Figure 2: Architecture of our **FEFA** and its multi-expert image fusion module.

pilot experiment with MAIRA-2: drop CL from CF+CL and CF+CL+PF cases, then evaluate the clinical accuracy of the generated reports across 14 observations labeled by CheXbert (Smit et al., 2020). The results are shown in Table 1, which confirms that removing lateral views from CF+CL cases actually boosts performance. This reveals a fusion flaw in MAIRA-2, where additional lateral images act as noise rather than helpful cues.

## 3.3 MULTI-EXPERT IMAGE FUSION

To solve this issue, we propose a multi-expert image fusion module and construct our **FEFA** as illustrated in Figure 2. Specifically, we employ three distinct expert networks to process the images across all four input types. Let $E_i^{fro}, E_i^{lat}, E_i^{pri} \in \mathbb{R}^{l \times d}$ denote the feature sequences of $x_i^{fro}, x_i^{lat}, x_i^{pri}$ after the image encoder, respectively. For current images, we enhance the CF features using self-attention, or fuse the CF and CL features through cross-attention, to obtain the fused image features $H_{cur} \in \mathbb{R}^{l \times d}$. If only current images are provided (i.e., CF or CF+CL), $H_{cur}$ is directly output as the final fused image features $H$. Formally,

$$\text{Expert 1 for CF}: H = H_{cur} = \text{Norm}\left(\text{Self-Att}(E_i^{fro}) + E_i^{fro}\right) \qquad (1)$$

$$\text{Expert 2 for CF+CL}: H = H_{cur} = \text{Norm}\left(\text{Cross-Att}(E_i^{fro}, E_i^{lat}, E_i^{lat}) + E_i^{fro}\right) \qquad (2)$$

When $E_i^{pri}$ is available, a gated network continues the fusion by adaptively mixing current and prior features. Specifically, we first compute the difference $H_\Delta$ between $H_{cur}$ and $E_i^{pri}$ as:

$$H_\Delta = \text{FFN}_1(H_{cur} - E_i^{pri}) \qquad (3)$$

Then, a dynamic fusion ratio $gate$ is learned based on the concatenation of $H_{cur}$ and $H_\Delta$.

$$gate = \text{Sigmoid}\left(\text{FFN}_2\left([H_{cur} : H_\Delta]\right)\right) \qquad (4)$$

After that, the weighted average of $H_{cur}$ and $H_\Delta$ is output as the final fused image feature $H$.

$$\text{Expert 3 for CF+PF \& CF+CL+PF}: H = \text{Norm}\left(gate \cdot H_{cur} + (1 - gate) \cdot H_\Delta\right) \qquad (5)$$

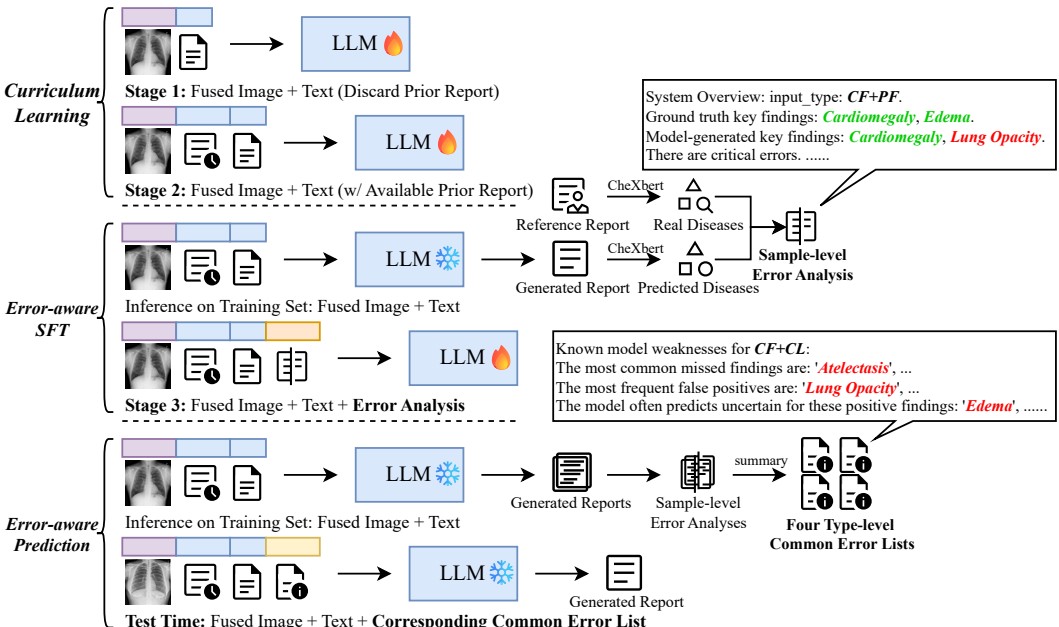

Figure 3: Architecture of our multi-stage training framework. When training the LLM, the multi-expert fusion module and the adapter are always be fine-tuned together.

With the proposed fusion module, *we transform the original input-dependent variable-length image features into a compact, fixed-length representation*. The shortened input feature sequence significantly reduces memory and computational requirements, speeding up both training and inference. Furthermore, the downstream LLM now faces only two input types, differing only in whether prior reports are provided (see Figure 2, right). In other words, image features now sit at fixed positions and always span the same token length. We posit that this deterministic layout eases learning, especially in our low-resource scenario. Some supporting analyses are provided in Section 4.5.

## 3.4 CURRICULUM LEARNING

During low-resource training, we occasionally observe a notable phenomenon: while the overall performance improves, the performance on samples without longitudinal information exhibits a decline. We hypothesize that this suggests the model's over-reliance on prior reports, which leads to the neglect of more critical current images. To teach model the order of priority across multiple views, a two-stage curriculum learning strategy is employed as shown in Figure 3.

In stage 1, we discard all prior reports available from each sample, constraining the model to generate reports primarily based on image information, especially the current images. The training objective is formalized as follows.

$$\text{Objective in Stage 1} : F_{\theta_1}(x_i^{fro}, x_i^{lat}, z_i, x_i^{pri}) \rightarrow y_i \tag{6}$$

In stage 2, we restore the prior reports in the samples, training the model to exploit longitudinal cues while preserving the image-centric skills acquired in stage 1. Formally,

$$\text{Objective in Stage 2} : F_{\theta_2}(x_i^{fro}, x_i^{lat}, z_i, x_i^{pri}, y_i^{pri}) \rightarrow y_i \tag{7}$$

## 3.5 ERROR-FEEDBACK AUGMENTATION

To better adapt to low-resource settings, we argue that more supervisory signals should be extracted from the limited annotated data. Inspired by Self-Refine (Madaan et al., 2023), we also aim to enable the model to perform self-correction based on feedback. However, the approach cannot be directly replicated. This is because, after fine-tuning for the RRG task, the model loses the multi-task capabilities inherent in the original LLM. During inference, it can neither evaluate reports, generate feedback, nor refine its outputs based on feedback.

This limitation, however, motivates us to explore whether the instruction-following capability inherent in LLMs can be leveraged for self-correction during the RRG training itself. For example, if we augment the prompt with the error types the model typically makes and explicitly warn it to avoid them, a well-trained general LLM can often produce better output. Although this implicit "reflection" may not originate from explicitly supervised data, we can construct tailored pairs during task-specific training to acquire or enhance this ability.

Based on this insight, we propose an error-feedback augmentation strategy. By injecting potential errors into the input, we prompt the model to notice and fix them towards the target RRG task. We refer to the training and inference stages under this strategy as error-aware Supervised Fine-Tuning (SFT) and error-aware prediction, respectively, with the process illustrated in Figure 3.

**Error-aware SFT.** To obtain error feedback, we first use our stage-2 model to generate reports for all training samples. Then, both generated and reference reports are labeled with CheXbert (Smit et al., 2020), obtaining their classification results on 14 clinical observations. For each sample $x_i$, we fill the label discrepancies into a predefined prompt to construct its error analysis $e_i$ (a full instance is shown in Figure 12). After that, the sample-level error analysis is incorporated into the input as feedback, initiating a new stage-3 training. Formally,

$$\text{Objective in Stage 3}: F_{\theta_3}(x_i^{fro}, x_i^{lat}, z_i, x_i^{pri}, y_i^{pri}, e_i) \rightarrow y_i \tag{8}$$

**Error-aware Prediction.** At test time, although sample-level error analysis is unavailable, we can still prompt the model with common errors identified during training. Specifically, we first obtain error analyses for the stage-3 model on the training set, and then extract the most frequent errors for each of the four input types. These errors are inserted into another predefined prompt to create four type-level common error lists (a full instance is shown in Figure 13). For a given test sample $x$, we incorporate its corresponding common error list $c_j$ into the input, serving as coarse-grained feedback to the model. Formally,

$$\text{Prediction at Test Time}: F_{\theta_3}(x^{fro}, x^{lat}, z, x^{pri}, y^{pri}, c_j) \rightarrow y_{pred} \tag{9}$$

where $j \in \{\text{CF}, \text{CF+CL}, \text{CF+PF}, \text{CF+CL+PF}\}$.

In summary, rather than post-editing a finished report, we train the model to proactively focus on potential errors before generating the report, thereby preventing them. To this end, we utilize error analysis and common error lists to reflect issues from the previous stage, prompting the model to correct them in the current training or inference. Despite the absence of explicit error-correction supervision or dedicated training process, our strategy still enables implicit self-correction within the original task training.

## 4 EXPERIMENTATION

### 4.1 EXPERIMENT SETTINGS

**Datasets.** MIMIC-CXR (Johnson et al., 2019) is a large-scale public dataset comprising paired chest X-rays and radiology reports. Each pair contains a varying number of images and all pairs for a patient are organized chronologically. IU X-ray (Demner-Fushman et al., 2016) is another small dataset without longitudinal data. We use the entire IU X-ray only for testing and establish our low-resource setup on MIMIC-CXR. Following MAIRA-2 (Bannur et al., 2024), we treat the "FINDINGS" section in radiology reports as the reference, filter the official split, and obtain 175,906 samples for training and 2,461 samples for testing. We then sample 5k, 15k, and 30k instances from the training set while maintaining a balanced distribution across four input types, yielding three low-resource training splits. Our models are trained separately on these three small splits and evaluated on the same test set.

**Baselines.** We primarily compare our FEFA with the reproduced MAIRA-2 trained on the same low-resource splits, as well as SRFE (Li et al., 2024) and RAMT (Zhang et al., 2024) that trained with 25% of MIMIC-CXR. Some recent high-resource approaches (Liang et al., 2024; Hou et al., 2024; Yin et al., 2025; Huang et al., 2025b;a) are also included as references. The official MAIRA-2, trained with full MIMIC-CXR and additional (including private) data, is treated as an *upper bound*.

**Evaluation Metrics.** Following most prior works, we evaluate the effectiveness of our FEFA using both NLG and Clinical Efficacy (CE) metrics. Considering that standard NLG metrics may not

Table 2: Performance comparisons between competitive baselines and SOTA MAIRA-2 on MIMIC-CXR. * represents our re-evaluation using the official released checkpoint, while others are derived from original papers. CE metrics here generally refer to the micro-average scores of CX-14 (u→n).

| Training Scale | Venue | Model | NLG Metrics | | | CE Metrics | | |
|---|---|---|---|---|---|---|---|---|
| | | | B-4 | MTR | R-L | P | R | F |
| 25% of MIMIC-CXR (40k around) | MICCAI'24 | SRFE (Li et al., 2024) | 10.4 | 13.8 | 27.7 | 32.5 | 21.4 | 21.5 |
| | TMM'24 | RAMT (Zhang et al., 2024) | 10.6 | 15.3 | 28.9 | 36.2 | 30.4 | 30.9 |
| Full MIMIC-CXR (160k around) | ACM MM'24 | DCG (Liang et al., 2024) | 12.6 | 16.2 | 29.5 | 44.1 | 41.4 | 40.4 |
| | EMNLP'24 | ICON (Hou et al., 2024) | 12.6 | 17.0 | 28.7 | 44.5 | 50.5 | 46.4 |
| | COLING'25 | KIA (Yin et al., 2025) | 13.8 | 16.7 | 30.7 | 50.4 | 42.5 | 46.1 |
| | AAAI'25 | DAMPER (Huang et al., 2025b) | 19.3 | 28.9 | 30.1 | 51.2 | 47.3 | 50.7 |
| | NAACL'25 | DDGIP (Huang et al., 2025a) | 11.7 | 16.7 | 28.8 | 50.8 | 52.9 | 51.8 |
| MIMIC-CXR and more (400k+) | ArXiv'24 | MAIRA-2 | 23.1 | 41.7 | 38.4 | - | - | 58.1 |
| | | MAIRA-2* | 13.9 | 34.9 | 32.1 | 59.3 | 53.2 | 56.1 |

Table 3: Performance comparisons with SOTA MAIRA-2 on varing low-resource training sets on MIMIC-CXR. The best results are highlighted in blue .

| Training Scale | Model | NLG Metrics | | | CX-14 (u→p) | | CX-14 (u→n) | |
|---|---|---|---|---|---|---|---|---|
| | | B-4 | MTR | R-L | Mac-F1 | Mic-F1 | Mac-F1 | Mic-F1 |
| 5k | MAIRA-2 | 4.67 | 23.44 | 20.28 | 25.96 | 36.99 | 22.27 | 35.98 |
| | FEFA | 5.64 | 27.34 | 20.93 | 35.20 | 47.59 | 30.36 | 47.27 |
| 15k | MAIRA-2 | 5.02 | 24.61 | 21.19 | 27.76 | 39.31 | 24.18 | 39.15 |
| | FEFA | 6.54 | 29.65 | 22.32 | 38.15 | 51.08 | 33.32 | 51.17 |
| 30k | MAIRA-2 | 6.36 | 27.60 | 22.55 | 33.50 | 46.72 | 28.86 | 46.80 |
| | FEFA | 6.84 | 30.66 | 22.69 | 39.45 | 52.21 | 34.51 | 53.18 |
| 400k+ | MAIRA-2 | 13.89 | 34.93 | 32.07 | 43.07 | 55.70 | 39.86 | 56.10 |

fully reflect clinical accuracy, we report BLEU-4 (B-4), METEOR (MTR), and ROUGE-L (R-L) for completeness, but focus on the CE scores derived via CheXbert (Smit et al., 2020). Specifically, after labeling reports with CheXbert to obtain multi-class labels for 14 clinical observations (CX-14), we map each "uncertain" label to both positive (u→p) and negative (u→n), then compute macro- and micro-average F1 for each mapping scheme. All four F1 scores are reported.

## 4.2 MAIN RESULTS

We primarily report the performance on MIMIC-CXR here. The results on IU X-ray which follow the same trend are provided in A.2. Since previous studies typically report CE metrics under a single mapping scheme, we consolidate these results in Table 2 for clarity. Performance comparisons in our low-resource settings are presented in Table 3, which demonstrates the superiority of our FEFA on the low-resource RRG task.

From the tables, we can observe that: **(i)** Across all three training scales, our FEFA consistently outperforms the corresponding MAIRA-2 on both NLG and CX-14 metrics, with performance scaling as the training data increase. **(ii)** On the CX-14 metrics, our FEFA trained on 5k samples surpasses SRFE and RAMT which used more data. With 30k reports, our FEFA outperforms all competitive baselines in Table 2, achieving approximately **94%** of the official MAIRA-2's micro-average F1 scores. **(iii)** Although our FEFA underperforms other baselines on NLG metrics, this does not indicate inferiority. As the example in Figures 5 and more examples in A.7 illustrate, our generated reports exhibit comparable fluency to the reference reports. The gap merely reflects lexical deviations, while clinical accuracy remains more important.

Additionally, we demonstrate the robustness and scalability of FEFA in A.4. It performs consistently well under varying sampling strategies (A.4.1), different LLM backbones (A.4.2), expanded training scales (A.4.3), and even shows promise at integrating prior lateral views during inference (A.4.4).

## 4.3 ABLATION STUDY

Our improvements to MAIRA-2 operate at three different levels: the internal model structure (multi-expert image fusion), the standard training phase (curriculum learning), and the post-training stage

Table 4: Ablation studies of different components. Modules we proposed or employed are added to MAIRA-2 step by step, until it becomes our FEFA. The best results and second best results are highlighted in blue and red , respectively.

| Training Scale | Model | Multi-expert Image Fusion | Curriculum Learning | Error-aware SFT | Error-aware Prediction | NLG Metrics | | | CX-14 (u→p) | | CX-14 (u→n) | |
|---|---|---|---|---|---|---|---|---|---|---|---|---|
| | | | | | | B-4 | MTR | R-L | Mac-F1 | Mic-F1 | Mac-F1 | Mic-F1 |
| 5k | MAIRA-2 | | | | | 4.67 | 23.44 | 20.28 | 25.96 | 36.99 | 22.27 | 35.98 |
| | (a) | ✓ | | | | 4.42 | 25.32 | 19.53 | 28.27 | 40.04 | 23.73 | 39.89 |
| | (b) | ✓ | ✓ | | | 4.31 | 26.21 | 19.84 | 31.99 | 44.62 | 27.40 | 44.43 |
| | (c) | ✓ | ✓ | ✓ | | 5.66 | 27.65 | 21.16 | 31.62 | 44.31 | 27.56 | 44.57 |
| | FEFA | ✓ | ✓ | ✓ | ✓ | 5.64 | 27.34 | 20.93 | 35.20 | 47.59 | 30.36 | 47.27 |
| 15k | MAIRA-2 | | | | | 5.02 | 24.61 | 21.19 | 27.76 | 39.31 | 24.18 | 39.15 |
| | (a) | ✓ | | | | 4.56 | 25.99 | 19.05 | 32.45 | 44.86 | 26.95 | 44.60 |
| | (b) | ✓ | ✓ | | | 5.67 | 28.85 | 21.31 | 37.50 | 49.66 | 32.84 | 50.28 |
| | (c) | ✓ | ✓ | ✓ | | 6.52 | 30.08 | 22.50 | 36.70 | 48.95 | 31.54 | 48.95 |
| | FEFA | ✓ | ✓ | ✓ | ✓ | 6.54 | 29.65 | 22.32 | 38.15 | 51.08 | 33.32 | 51.17 |
| 30k | MAIRA-2 | | | | | 6.36 | 27.60 | 22.55 | 33.50 | 46.72 | 28.86 | 46.80 |
| | (a) | ✓ | | | | 5.23 | 27.26 | 20.59 | 34.69 | 46.78 | 30.48 | 47.29 |
| | (b) | ✓ | ✓ | | | 5.90 | 28.28 | 21.75 | 37.83 | 50.90 | 32.98 | 51.37 |
| | (c) | ✓ | ✓ | ✓ | | 6.12 | 28.49 | 21.93 | 37.49 | 50.12 | 32.71 | 50.71 |
| | FEFA | ✓ | ✓ | ✓ | ✓ | 6.84 | 30.66 | 22.69 | 38.93 | 52.24 | 34.51 | 53.18 |

Table 5: Results of the same pilot experiment on our models. (a) is a MAIRA-2 equipped with our multi-expert image fusion module, identical to that in Table 4.

| Model | Input Type | Training Scale: 5k CX-14 (u→n) | | Training Scale: 15k CX-14 (u→n) | | Training Scale: 30k CX-14 (u→n) | |
|---|---|---|---|---|---|---|---|
| | | Mac-F1 | Mic-F1 | Mac-F1 | Mic-F1 | Mac-F1 | Mic-F1 |
| (a) | CF+CL | **18.95** | **25.74** | **20.81** | **32.71** | **23.04** | **36.65** |
| | discard CL | 14.16 | 24.36 | 18.78 | 28.97 | 22.03 | 33.72 |
| FEFA | CF+CL | **24.05** | **37.60** | **30.54** | **38.54** | **30.96** | **40.06** |
| | discard CL | 21.41 | 30.28 | 22.75 | 35.37 | 24.22 | 36.02 |

(error-feedback augmentation). By gradually integrating these modules and strategies into MAIRA-2, we construct the FEFA. To elucidate the contribution of each component and their cumulative effects, we present the performance of models with different modules incrementally incorporated in Table 4, and additional ablation analysis on our final FEFA in A.3.

**Effect of Multi-expert Image Fusion.** In Table 4, (a) denotes a MAIRA-2 equipped with our multi-expert image fusion module, and trained only through stage 2. Compared to MAIRA-2, while some NLG metrics show a slight decline, (a) achieves substantial improvements across all CX-14 metrics. This indicates that our proposed fusion mechanism enables the model to better focus on clinically relevant regions within the images. Moreover, thanks to the compact fused features, (a) requires only half the training time of MAIRA-2, making it friendly for compute-limited environments.

To verify that our modification fixes MAIRA-2's fusion flaw, we repeat the pilot experiment on our models. The results in Table 5 show that our ablated model (a) consistently gains performance when provided with extra lateral images. This advantage of effectively utilizing lateral cues persists until our final FEFA.

**Effect of Curriculum Learning.** Compared to (a), (b) improves on nearly all metrics, with CX-14 gains up to 5%. This underscores both the importance and benefits of progressively integrating multi-view information in low-resource settings.

**Effect of Error-feedback Augmentation.** Compared with (b), (c) improves on NLG metrics while maintaining comparable or slightly lower CX-14 scores. This suggests that implicitly training the model to focus on its own errors has a minimal impact on its primary task performance. Finally, compared to (b) and (c), our FEFA which also receives feedback at test time achieves further significant gains on CX-14 metrics. This confirms that with our augmentation strategy, the model can indeed reflect on feedback to avoid potential errors and achieve implicit self-correction. To visually illustrate this advantage, we randomly select some samples and compare the reports generated by (b) and our final FEFA. The examples in Figures 5 and 11 demonstrate that our FEFA can outperform (b) in reducing both false positives and missed diagnoses.

Table 6: Performance gains of FEFA (trained with 30k samples) across each clinical observation in CX-14 when using errors of different frequencies as feedback during inference. "Obs." is short for observations and positive ΔAcc. are highlighted in green.

| 14 Clinical Observations Labeled by CheXbert | Acc. of FEFA w/o Error-aware Prediction | FEFA (w/ Most Frequent Errors) | | FEFA w/ Moderate Frequent Errors | | FEFA w/ Least Frequent Errors | |
|---|---|---|---|---|---|---|---|
| | | ΔAcc. | Obs. in Feedback | ΔAcc. | Obs. in Feedback | ΔAcc. | Obs. in Feedback |
| Enlarged Cardiomediastinum | 87.85 | 0.16% | ✓ | -1.99% | ✓ | -1.54% | |
| Cardiomegaly | 70.58 | -2.03% | ✓ | -1.95% | ✓ | -2.28% | |
| Lung Opacity | 62.82 | 1.14% | ✓ | 0.24% | | 0.20% | ✓ |
| Lung Lesion | 92.32 | 0.93% | | 1.06% | ✓ | 0.53% | ✓ |
| Edema | 79.20 | 1.30% | ✓ | 1.18% | ✓ | 0.08% | ✓ |
| Consolidation | 92.32 | 1.63% | | -0.08% | ✓ | 0.16% | ✓ |
| Pneumonia | 93.25 | 1.46% | ✓ | 1.38% | ✓ | 0.61% | ✓ |
| Atelectasis | 69.08 | 1.22% | ✓ | -1.95% | | 1.02% | |
| Pneumothorax | 96.46 | 1.46% | | 0.81% | ✓ | 1.02% | ✓ |
| Pleural Effusion | 76.68 | 1.58% | ✓ | 0.49% | ✓ | -0.45% | |
| Pleural Other | 95.73 | 0.65% | | 0.37% | ✓ | -0.20% | ✓ |
| Fracture | 94.80 | -0.41% | | -0.73% | ✓ | -1.10% | ✓ |
| Support Devices | 84.19 | -0.08% | | -1.26% | ✓ | -0.65% | |
| No Finding | 88.58 | 1.02% | ✓ | 3.01% | ✓ | 2.93% | |
| Avg. of All Obs. | 84.56 | 0.72% | | 0.04% | | 0.02% | |
| Avg. of Obs. in Feedback | - | **0.73%** | | **0.19%** | | **0.16%** | |
| Avg. of Obs. Not in Feedback | - | 0.70% | | -0.85% | | -0.16% | |

### 4.4 ANALYSIS ON SELF-CORRECTION ACHIEVED BY ERROR-FEEDBACK AUGMENTATION

To better illustrate the self-correction achieved by our error-feedback augmentation, we present the detailed performance across 14 clinical observations labeled by CheXbert under three feedback strategies in Table 6. Prior to inference, we quantify three categories of errors (misses, false positives, and uncertainties) on the entire training set (as shown in Figure 3). Then by default, we incorporate the four most frequent clinical observations for each error type into the feedback, allowing for potential overlaps between different categories. Additionally, we also explored utilizing moderately frequent (ranks 6–9 out of 14) and the four least frequent errors to construct feedback.

As shown in Table 6, the average accuracies across all observations surpass the non-feedback baseline, regardless of the frequency of errors we used. Most observations mentioned in feedback exhibit improved performance. Moreover, the average performance gains on observations in feedback are consistently positive and higher than that on observations which are not included. This clearly demonstrates both the effectiveness and robustness of our error-feedback augmentation approach.

From another perspective, we can also observe that leveraging more frequent errors as feedback leads to greater performance gains. When the top four frequent errors are used, 12 out of 14 observations show accuracy gains, whereas only 8 categories improve when moderately frequent or infrequent errors are utilized. Quantitatively, the average gains across all observations and those mentioned in feedback decrease as the error frequency drops. This matches intuition and supports the rationale for selecting the most frequent errors as feedback.

### 4.5 DISCUSSION: VISUALIZING THE SUPERIORITY OF OUR FUSED IMAGE BY UID THEORY

The above experimental analysis shows that compared to MAIRA-2, which uses simple concatenation to process image information, our fusion module utilizes lateral images more effectively and achieves better results under the same dataset size. Since the fused representation contains no more raw information than concatenation, the gains are unlikely due to the information volume. Therefore, we hypothesize that they arise because the fused inputs are easier for the LLM to interpret.

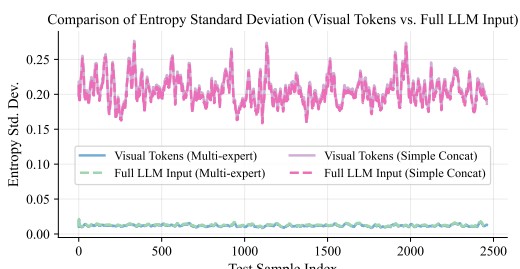

Figure 4: Token-wise entropy Std. Dev. between visual tokens and full LLM input (↓ uniformity).

To quantify this "understandability", we introduce the Uniform Information Density (UID) theory. The UID principle posits that production and comprehension are most efficient when information is distributed at a relatively even rate, avoiding sharp "peaks" (overload) and "troughs" (under-informative spans) (Jaeger, 2010; Meister et al., 2021). The cornerstone of UID theory is "Surprisal". For a linguistic unit $u$ (e.g., a word), its surprisal is the negative log conditional probability

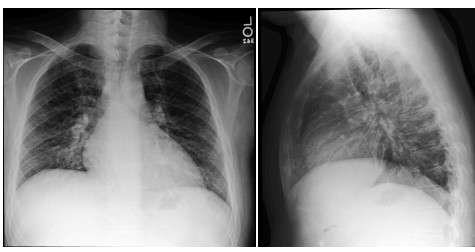

**Ground Truth:** Central pulmonary vascular congestion with moderate pulmonary edema. No focal consolidation, pneumothorax, or pleural effusion.

**w/o EFA:** Cardiomediastinal silhouette is stable. Mild pulmonary vascular congestion without frank interstitial edema or pleural effusion. There are mild bibasilar opacities, most likely atelectasis and not pneumonia. No focal consolidation to suggest infection.

**FEFA:** PA and lateral chest radiographs demonstrate a normal cardiomediastinal silhouette. There is mild pulmonary vascular congestion with minimal interstitial edema, without pleural effusion or pneumothorax. The lungs are otherwise clear.

Figure 5: A case of reports generated by FEFA w/ and w/o our Error-Feedback Augmentation (EFA). Findings corresponding to the ground-truth are annotated with the same color, while spans in red denote false positive observations.

given context: $I(u) = -\log(P(u|\text{context}))$. So low-probability, unpredictable events carry higher information (higher surprisal) (Clark et al., 2023). In our setting, vision features are projected into the text space and consumed by the LLM as a token sequence, making UID a natural analytical frame. Under this lens, concatenation inflates low-surprisal redundancy and dilutes high-surprisal, view-specific cues, yielding a signal with high variance in information density, which is inconsistent with UID's preference for stable, balanced signals (Meister et al., 2021). In contrast, multi-expert fusion performs pre-LLM selection or distillation that concentrates diagnostically meaningful high-surprisal evidence while smoothing low-surprisal regions, producing a more uniform input profile.

To verify our hypothesis, we analyze the within-sample dispersion of token entropy under two views: (i) the standard deviation (std. dev.) of entropy over visual tokens after fusion and (ii) the std. dev. of entropy over all input tokens before LLM. As shown in Figure 4, our method yields lower entropy dispersion than simple concatenation in both cases, indicating that a more uniform and UID-consistent signal is supplied to LLM, which can reduce its processing burden and facilitate better understanding. We conducted complementary analyses of attention behavior within visual fusion and LLM to further verify whether it conforms to the UID theory. Details are provided in A.5.

### 4.6 CASE STUDY

Figure 5 illustrates a real example of radiology report generation using our proposed FEFA, with and without the Error-Feedback Augmentation (EFA) mechanism. As visible in the figure, the model variant without EFA erroneously produces the phrase "mild bibasilar opacities," which aligns with the second type of hallucination summarized by our EFA strategy, namely Model-generated key findings (also outlined in Figure 3). In contrast, when EFA is incorporated (as Figure 3: "The most frequent false positives are..."), FEFA successfully avoids such inaccuracies and generates clinically relevant content consistent with the imaging findings. This improvement underscores the capacity of the EFA paradigm to mine richer supervisory signals from limited image-report pairs and enhance the factual reliability of automated generation. More cases can refer to Appendix.

## 5 CONCLUSION

To address data scarcity and challenges in fusing multi-view features for radiology report generation (RRG), we proposed FEFA, which combines a multi-expert image fusion module and an error-feedback augmentation strategy. Experiments on MIMIC-CXR showed FEFA achieves strong data efficiency—reaching 94% of the state-of-the-art clinical efficacy with only 8% of its supervised data, while outperforming competing methods. This work advances RRG in low-resource scenarios and provides a practical approach to leverage error insights for medical AI models with limited data.

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

# A APPENDIX

## A.1 IMPLEMENTATION DETAILS

We follow MARIA-2 for sample construction. For the prior report, only the "FINDINGS" section is retained, and "COMPARISON" is discarded from the auxiliary text. For studies lacking temporal input, we apply LLaVA-Rad MIMIC-CXR annotations (Chaves et al., 2024; Chaves et al.) to remove temporal mentions from the target texts.

Our model comprises a frozen image encoder, a learnable adapter over image features, a multi-expert fusion module, and an LLM. We use a re-trained Rad-DINO as the frozen vision encoder (Pérez-García et al., 2024) and initialize the language model with the weights of Vicuna-7B v1.5 (Chiang et al., 2023), which is the same as MAIRA-2. Training uses standard autoregressive cross-entropy loss (Graves, 2013) with LoRA fine-tuning for **4 epochs per stage**. We apply LoRA with rank = 8 and $\alpha = 32$, trainable modules = [q_proj, k_proj, v_proj]. We adopt AdamW with a cosine schedule, a 3% warm-up, and an initial learning rate of $2 \times 10^{-5}$ for Stages 1–2 and $1.5 \times 10^{-5}$ for Stage 3. Global batch sizes per stage are: {1, 1, 4} for the 5k subset; {2, 2, 16} for the 15k subset; and {4, 4, 32} for the 30k subset. For inference, we decode in bfloat16 precision for up to 256 tokens.

For inference, we decode in bfloat16 precision for up to 256 tokens. All experiments were performed on 2 × NVIDIA RTX 4090 GPUs. The complete prompt we constructed is shown in Figure 14.

## A.2 EXPERIMENTAL RESULTS ON IU X-RAY

IU X-ray (Demner-Fushman et al., 2016) is another small dataset without longitudinal data. Images in it may present in frontal alone or both frontal and lateral views per report. Following MAIRA-2, we only use IU X-ray for testing and obtain 3,307 valid samples after filtering.

Similar to experiments on MIMIC-CXR, we aggregate the results of baselines which adopt the same settings (trained on MIMIC-CXR and tested on IU X-ray) in Table 7. Performance comparisons in our low-resource settings are presented in Table 8.

Table 7: Performance comparisons between existing competitive baselines and SOTA on IU X-ray. * represents our re-evaluation using the official released checkpoint, while others are derived from original paper. CE metrics here generally refer to the micro-average scores of CX-14 (u→n).

| Training Scale | Venue | Model | NLG Metrics | | | CE Metrics | | |
|---|---|---|---|---|---|---|---|---|
| | | | B-4 | MTR | R-L | P | R | F |
| 25% of MIMIC-CXR (40k around) | MICCAI'24 | SRFE (Li et al., 2024) | 8.70 | 15.8 | 30.2 | 19.6 | 17.1 | 16.0 |
| Full MIMIC-CXR (160k around) | EMNLP'20 | R2Gen (Chen et al., 2020) | 5.90 | 13.1 | 25.3 | 14.1 | 13.6 | 13.6 |
| | AIM'23 | CVT2Dis (Nicolson et al., 2023) | 8.20 | 14.7 | 27.7 | 17.4 | 17.2 | 16.8 |
| | MIA'23 | M2KT (Yang et al., 2023) | 7.80 | 15.3 | 26.1 | 15.3 | 14.5 | 14.5 |
| | CVPR'23 | DCL (Li et al., 2023a) | 7.40 | 15.2 | 26.7 | 16.8 | 16.7 | 16.2 |
| | CVPR'23 | RGRG (Tanida et al., 2023) | 6.30 | 14.6 | 18.0 | 18.3 | 18.7 | 18.0 |
| | AAAI'24 | PromptMRG (Jin et al., 2024) | 9.80 | 16.0 | 28.1 | 21.3 | 22.9 | 21.1 |
| Full MIMIC-CXR and more (400k+) | ArXiv'24 | MAIRA-2 | 11.7 | - | 27.4 | - | - | 52.5 |
| | | MAIRA-2* | 8.29 | 33.7 | 28.3 | 50.6 | 54.7 | 52.6 |

Table 8: Performance comparisons with SOTA MAIRA-2 on varing low-resource training sets on IU X-ray. The best results are highlighted in blue .

| Training Scale | Model | NLG Metrics | | | CX-14 (u→p) | | CX-14 (u→n) | |
|---|---|---|---|---|---|---|---|---|
| | | B-4 | MTR | R-L | Mac-F1 | Mic-F1 | Mac-F1 | Mic-F1 |
| 5k | MAIRA-2 | 6.06 | 32.33 | 23.33 | 9.42 | 23.88 | 8.84 | 25.43 |
| | FEFA | 7.62 | 34.64 | 24.92 | 19.83 | 46.20 | 18.83 | 48.72 |
| 15k | MAIRA-2 | 6.50 | 32.45 | 23.93 | 13.67 | 35.45 | 12.48 | 38.15 |
| | FEFA | 7.33 | 33.87 | 25.27 | 22.66 | 46.76 | 20.75 | 49.41 |
| 30k | MAIRA-2 | 6.74 | 32.75 | 24.47 | 21.27 | 35.09 | 18.88 | 37.49 |
| | FEFA | 7.22 | 34.79 | 25.93 | 27.15 | 46.76 | 25.82 | 49.48 |
| 400k+ (Official) | MAIRA-2 | 8.29 | 33.67 | 28.30 | 31.29 | 50.44 | 29.34 | 52.56 |

From the tables we can see that our FEFA maintains its high performance on IU X-ray, too. Across all three training scales, it consistently outperforms the corresponding MAIRA-2 on both NLG and CX-14 metrics, with performance improving as the training data increase. On the CX-14 metrics, FEFA trained on 5k samples has already surpassed all competitive baselines in Table 7 which used more supervised data. Compared to official MAIRA-2 trained on more than 400k data, FEFA with 30k samples can achieve approximately **93%** of its micro-average F1 scores. These results collectively demonstrate the strong generalization capability of our model.

## A.3 ADDITIONAL ABLATION ANALYSIS

For a detailed analysis of component independence and their individual contributions to the final FEFA model, we present the performance of FEFA with each component isolated in Table 9.

The results show that removing any individual component leads to a significant drop in CX-14 performance; however, all ablated configurations still surpass the baseline MAIRA-2 across all metrics. This indicates that each of our proposed components is both independent and effective. Furthermore, we can observe that the multi-expert image fusion module contributes the most to the final performance. This is intuitively reasonable, because unlike other components which operate without altering the model structure, it replaces the naive image concatenation mechanism to generate a more compact and task-relevant representation, and serves as the cornerstone of our new architecture.

## A.4 EXPLORATION INTO THE ROBUSTNESS AND SCALABILITY OF FEFA

### A.4.1 RESULTS UNDER NON-UNIFORM SAMPLING STRATEGY

We adopt an input-type balanced sampling strategy to create our low-resource training splits mostly for convenience. Nevertheless, we also want to investigate the impact of sampling strategies on our FEFA's performance. To evaluate it, we randomly sample across the entire training set to construct

Table 9: Thorough ablation studies on MIMIC-CXR, which isolate each component we proposed or utilized from the final FEFA. "EFA", "CL", "MEF" are short for error-feedback augmentation, curriculum learning, and multi-expert image fusion, respectively. "FEFA w/o CL & EFA" and "FEFA w/o EFA" have also been shown in Table 4 (i.e., variants (a) and (b)).

| Training Scale | Model | Multi-expert Image Fusion | Curriculum Learning | Error-feedback Augmentation | NLG Metrics B-4 | NLG Metrics MTR | NLG Metrics R-L | CX-14 (u→p) Mac-F1 | CX-14 (u→p) Mic-F1 | CX-14 (u→n) Mac-F1 | CX-14 (u→n) Mic-F1 |
|---|---|---|---|---|---|---|---|---|---|---|---|
| | **FEFA** | ✓ | ✓ | ✓ | 5.64 | **27.34** | 20.93 | **35.20** | **47.59** | **30.36** | **47.27** |
| | FEFA w/o EFA | ✓ | ✓ | | 4.31 | 26.21 | 19.84 | 31.99 | 44.62 | 27.40 | 44.43 |
| | FEFA w/o CL | ✓ | | ✓ | **5.79** | 26.44 | **21.26** | 30.54 | 40.15 | 26.13 | 38.77 |
| 5k | FEFA w/o MEF | | ✓ | ✓ | 5.34 | 26.56 | 20.49 | 28.28 | 39.46 | 22.88 | 38.00 |
| | FEFA w/o CL & EFA | ✓ | | | 4.42 | 25.32 | 19.53 | 28.27 | 40.04 | 23.73 | 39.89 |
| | FEFA w/o MEF & EFA | | ✓ | | 5.43 | 25.70 | 20.72 | 27.44 | 38.47 | 22.78 | 37.35 |
| | FEFA w/o MEF & CL | | | ✓ | 4.84 | 26.35 | 20.06 | 26.64 | 37.49 | 22.70 | 36.72 |
| | MAIRA-2 | | | | 4.67 | 23.44 | 20.28 | 25.96 | 36.99 | 22.27 | 35.98 |

Table 10: Performance comparisons under different sampling strategies of FEFA on MIMIC-CXR.

| Training Scale | Sampling Strategy | NLG Metrics B-4 | NLG Metrics MTR | NLG Metrics R-L | CX-14 (u→p) Mac-F1 | CX-14 (u→p) Mic-F1 | CX-14 (u→n) Mac-F1 | CX-14 (u→n) Mic-F1 |
|---|---|---|---|---|---|---|---|---|
| 15k | uniform (equal for each input type, **default**) | 6.54 | 29.65 | 22.32 | **38.15** | 51.08 | **33.32** | **51.17** |
| | non-uniform (randomly in the training set) | **7.24** | **30.51** | **23.07** | 38.05 | **51.25** | 31.55 | 50.12 |

a new 15k spilt (ratio of CF: 9%, CF+CF: 27%, CF+PF: 32%, CF+CL+PF: 32%), and retrain our FEFA on it.

As the results shown in Table 10, our model trained with this new imbalanced split achieves performance statistically comparable to that on a balanced one. Meanwhile, we can reasonably infer that at larger training scales (e.g., 30k or more), the impact of sampling variance would be more negligible. This confirms that the reported efficiency gains stem from the proposed architecture, rather than the data sampling strategy, and strongly demonstrates the robustness of our FEFA model.

### A.4.2 RESULTS WITH VARYING LLM BACKBONES

Our FEFA is designed to be model-agnostic so it is theoretically applicable to any multimodal LLM (MLLM) that follows the same architecture with varying LLMs and vision encoders. To empirically validate this generality, we replace the original Vicuna-7B v1.5 we used to two SOTA general-purpose LLMs from Microsoft: Phi-3-mini (Abdin et al., 2024) and Phi-4-mini (Abouelenin et al., 2025), and reproduce the comparisons between our FEFA and MAIRA-2. Since the vision encoder is always frozen in our FEFA, we choose to keep it fixed this time.

The results are provided in Table 11, which remain fully consistent with that in the main body of the paper. No matter which LLM is integrated, our FEFA always outperforms MAIRA-2 across all metrics. Given that both Phi-3 and Phi-4 are compact 3.8B parameter models, it is understandable that their peak performance does not reach that of our larger Vicuna model. These results serve as compelling evidence for the strong robustness of our approach.

Table 11: Performance comparisons with SOTA MAIRA-2 on different LLM backbones on MIMIC-CXR. The best results are highlighted in blue.

| Training Scale | Model | NLG Metrics B-4 | NLG Metrics MTR | NLG Metrics R-L | CX-14 (u→p) Mac-F1 | CX-14 (u→p) Mic-F1 | CX-14 (u→n) Mac-F1 | CX-14 (u→n) Mic-F1 |
|---|---|---|---|---|---|---|---|---|
| | MAIRA-2 (w/ Vicuna-7B v1.5) | 6.36 | 27.60 | 22.55 | 33.50 | 46.72 | 28.86 | 46.80 |
| | **FEFA** (w/ Vicuna-7B v1.5) | 6.84 | 30.66 | 22.69 | 38.93 | 52.24 | 34.51 | 53.18 |
| 30k | MAIRA-2 w/ Phi-3-mini (Abdin et al., 2024) | 4.11 | 24.88 | 17.19 | 16.80 | 28.86 | 12.48 | 25.45 |
| | **FEFA** w/ Phi-3-mini (Abdin et al., 2024) | 4.45 | 27.59 | 17.23 | 30.21 | 42.38 | 26.86 | 42.49 |
| | MAIRA-2 w/ Phi-4-mini (Abouelenin et al., 2025) | 2.14 | 19.55 | 12.20 | 15.10 | 27.10 | 12.80 | 26.54 |
| | **FEFA** w/ Phi-4-mini (Abouelenin et al., 2025) | 5.53 | 31.12 | 19.24 | 35.17 | 47.93 | 31.59 | 48.90 |

Table 12: Performance of **FEFA** on MIMIC-CXR trained with 60k samples. CX-5 represents the classification performance for five specific diseases (Atelectasis, Cardiomegaly, Consolidation, Edema, and Pleural Effusion) within CX-14. "%" indicates the ratio relative to the performance of official MAIRA-2. Values above 95% are highlighted in bold.

| Training Scale | Model | NLG Metrics | | | CX-14 (u→p) | | CX-14 (u→n) | | CX-5 (u→p) | | CX-5 (u→n) | |
|---|---|---|---|---|---|---|---|---|---|---|---|---|
| | | B-4 | MTR | R-L | Mac-F1 | Mic-F1 | Mac-F1 | Mic-F1 | Mac-F1 | Mic-F1 | Mac-F1 | Mic-F1 |
| 30k | **FEFA** | 6.84 | 30.66 | 22.69 | 38.93 | 52.24 | 34.51 | 53.18 | 47.98 | 54.83 | 42.96 | 52.72 |
| | **%** | 49.2% | 87.8% | 70.8% | 90.4% | 93.8% | 86.6% | 94.8% | 93.5% | 93.3% | 91.5% | 93.5% |
| 60k | **FEFA** | 10.13 | 31.69 | 28.02 | 41.71 | 53.88 | 37.25 | 53.55 | 50.62 | 57.31 | 45.43 | 54.15 |
| | **%** | 72.9% | 90.7% | 87.4% | **96.8%** | **96.7%** | 93.5% | **95.5%** | **98.6%** | **97.5%** | **96.8%** | **96.1%** |
| 400k+ (Official) | MAIRA-2 | 13.89 | 34.93 | 32.07 | 43.07 | 55.70 | 39.86 | 56.10 | 51.34 | 58.77 | 46.93 | 56.37 |

### A.4.3 RESULTS AT LARGER TRAINING SCALES

Our study primarily targets the low-resource regime, which reflects the practical constraints of medical applications. Nevertheless, we also want to examine the scalability of **FEFA** when more samples are available. To realize it, we conducted an additional experiment by doubling the training data from 30k to 60k, and provided CX-5 metrics for a more detailed evaluation.

The results shown in Table 12 exhibit a clear positive scaling trend. With 30k training samples, our **FEFA** has already achieved 94% of the official MAIRA-2's micro-average F1 scores, while its macro-F1 scores have not yet matched this level. With 60k training samples, performance further increases to 96% on CX-14 and 98% on CX-5, indicating continued improvements with more data. Furthermore, ratios on nearly all CX-14 metrics exceed 95%, demonstrating consistent high performance across the board. Notable gains are also observed on the NLG metrics. These results show no evidence of saturation and suggest that our model continues to benefit from additional supervision.

### A.4.4 RESULTS WHEN INTEGRATING PRIOR LATERAL VIEWS AT INFERENCE TIME

To ensure a fair comparison with MAIRA-2, we followed it to only use the current frontal (CF), current lateral (CL), and prior frontal (PF) views as visual inputs. Nevertheless, since the expert 2 in our proposed multi-expert fusion module captures cross-view interactions (CF&CL) and expert 3 models longitudinal discrepancies (CF&PF), we believe that our model also has the capability to incorporate additional prior lateral (PL) views.

To preliminarily validate this hypothesis, we conducted zero-shot inference experiments using the frozen **FEFA** on two types of test samples containing PL views. The results are shown in Table 13.

For samples containing PF views (left in Table 13), we reuse the spatial expert 2 to fuse PF and PL images. Then, the "enhanced prior features" are interacted with current features in the temporal expert 3. Given that the **FEFA** model was never trained to incorporate fused prior features, the minor performance drop is not only acceptable but also indicative of a certain level of inherent robustness.

For samples without PF views (right in Table 13), we apply the temporal expert 3 on CL and PL to obtain the "enhanced lateral features" and then fuse it with CF in the spatial expert 2. From the table we can see that, the inclusion of PL information yields a significant performance improvement on this type of samples. The result not only confirms that the temporal reasoning mechanism in expert 3 could generalize to prior lateral views even without explicit supervision, but also demonstrates the potential of our **FEFA** to effectively integrate additional images or views. We expect to further strengthen this ability via dedicated training in future work.

Table 13: CX-14 comparisons when inputting prior lateral images (PL) to **FEFA** (trained with 30k samples) during inference on MIMIC-CXR. "[+CL]" means current lateral images (CL) are optional.

| Input Type | CX-14 (u→n) | | Input Type | CX-14 (u→n) | |
|---|---|---|---|---|---|
| | Mac-F1 | Mic-F1 | | Mac-F1 | Mic-F1 |
| CF[+CL]+PF (w/ unused PL) | **32.48** | **48.50** | CF+CL (w/ unused PL) | 24.78 | 50.67 |
| CF[+CL]+PF+PL | 32.35 | 47.41 | CF+CL+PL | **28.30** | **56.41** |

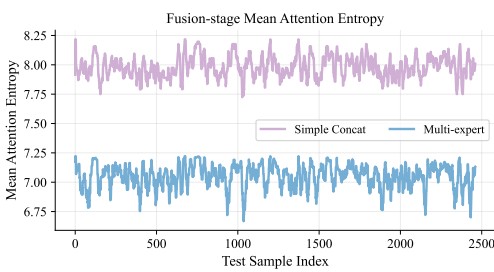 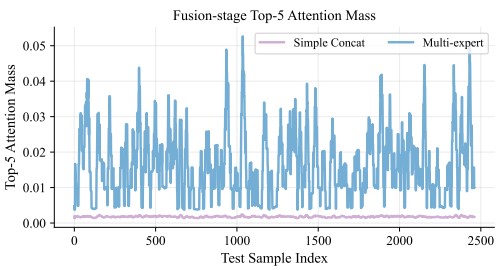

(a) Fusion-stage mean attention entropy (↓ selective)  (b) Fusion-stage Top-5 mass (↑ sparse focus)

Figure 6: Pre-LLM Sparsification at the Fusion Stage

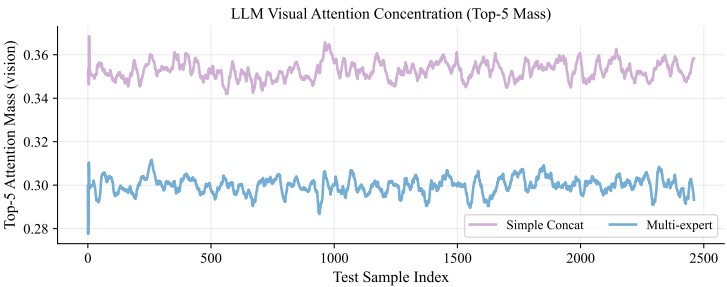

Figure 7: Visual Top-5 mass within LLM self-attention (↓ less over-concentration)

### A.5 VISUALIZING THE BEHAVIOR OF OUR FUSION MODULE AND LLM TO PROVE UID

The central criterion for UID theory is the uniform information density, which has been directly illustrated in Figure 4. Beyond Figure 4, we further probe (i) attention behavior during visual fusion and (ii) intra-LLM self-attention, summarized in Figures 6 and 7.

**Fusion-stage Diagnostics.** Figure 6 compares our multi-expert fusion with simple concatenation in the process of processing image information. Figure 6 (a) shows the mean entropy of the attention distribution of visual tokens within each sample during the fusion process. High entropy means that the attention is more dispersed and "averagely" pays attention to all information in the visual area. Low entropy means that the attention is concentrated on a few visual tokens. Figure 6 (b) reports the Top-5 mass, i.e., the summed of the attention masses of the top-5 visual tokens in the attention distribution of all tokens (again measured within-sample over visual tokens). A higher Top-5 mass reflects sparser and more targeted attention, suggesting that the fusion process filters information in a more selective way.

In summary, these diagnostics indicate that our multi-expert module performs pre-LLM sparsification: it extracts a compact signal from the long, noisy raw visual inputs while filtering out redundant noise, thereby preparing a shorter, more uniform, UID-consistent information stream for the LLM and reducing its processing burden.

**Intra-LLM Attention Behavior.** Figure 7 measures the Top-5 attention mass within the visual segment of the full sequence. Our method produces a lower Top-5 concentration compared to naive concatenation. This shows that the LLM does not over-focus on a handful of tokens but distributes attention more evenly, avoiding bottlenecks. This is consistent with our previous analysis that our method provides uniform and stable information for LLM. In contrast, concatenation produces disproportionately high Top-5 mass, indicating unhealthy over-concentration, which is a clear violation of UID.

**Suprisal Analysis.** A key motivation for applying UID theory in our study lies in its foundational concept: surprisal. Formally, surprisal is defined as the negative logarithm of the conditional probability of an event given its context (see Section 4.5 for detailed formulation). Under this definition, low-probability or unpredictable events carry higher information content and thus exhibit greater

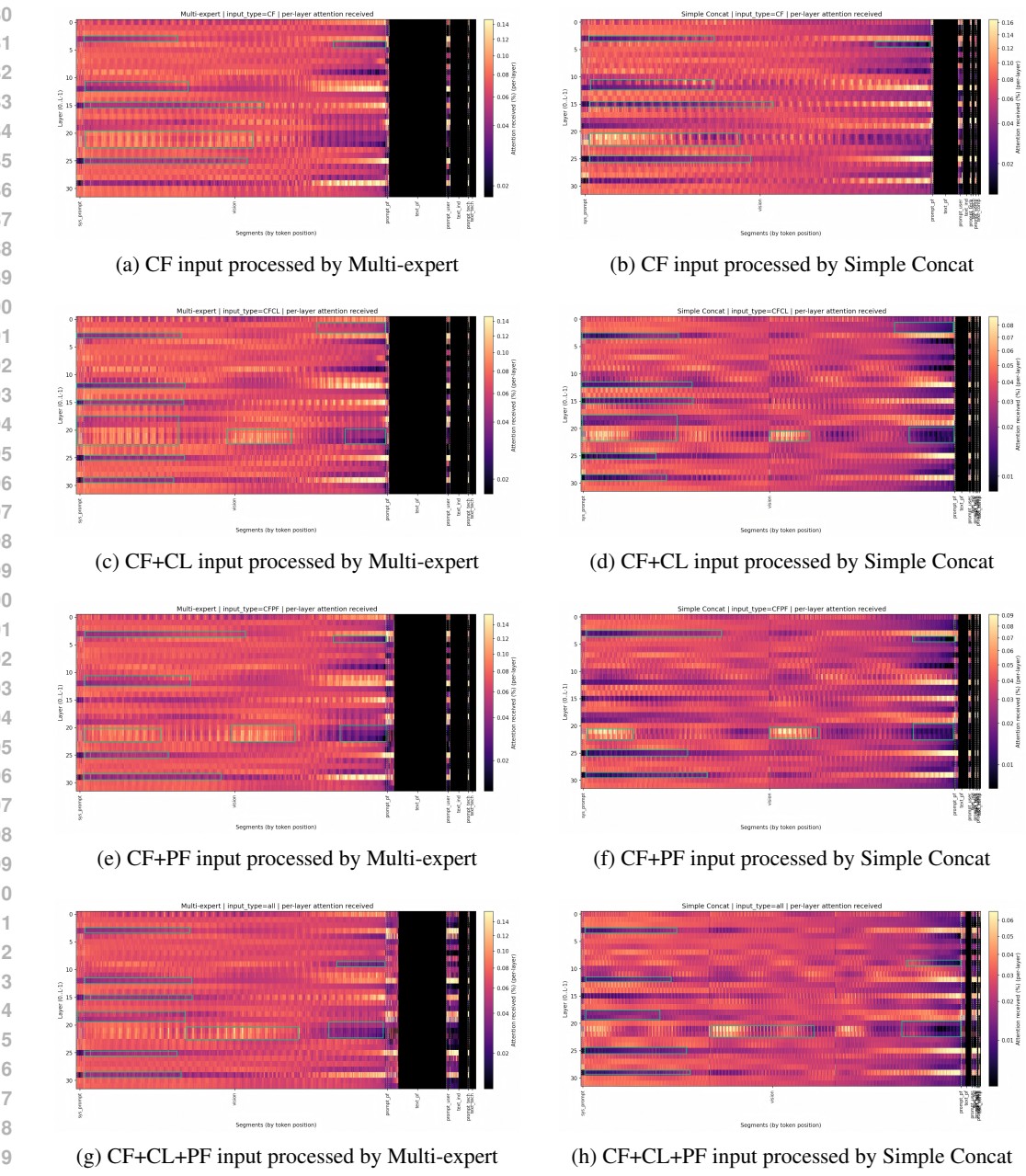

(a) CF input processed by Multi-expert

(b) CF input processed by Simple Concat

(c) CF+CL input processed by Multi-expert

(d) CF+CL input processed by Simple Concat

(e) CF+PF input processed by Multi-expert

(f) CF+PF input processed by Simple Concat

(g) CF+CL+PF input processed by Multi-expert

(h) CF+CL+PF input processed by Simple Concat

Figure 8: Layer-wise self-attention heatmaps within the LLM, showing how keys derived from four input types behave under two fusion strategies.

surprisal (Clark et al., 2023). This property naturally aligns with the characteristics of lesions in medical images, which are rare, informative, and inherently high-surprisal events.

Our goal in this analysis is to assess whether the information fed into the LLM conforms to UID principles. A lower variance in surprisal across the input sequence implies a more uniform information distribution, and stronger adherence to UID.

However, conventional surprisal estimation methods rely on perplexity, which computes surprisal based on generated outputs. This metric is incompatible with our setting because, during fusion, the LLM operates as a receiver rather than an autoregressive generator. Thus, its output probability distribution is not available for computing perplexity. To address this limitation, we return to the fundamental definition of surprisal: events with lower conditional probability carry higher sur-

prisal and thus convey more information. In Transformer-based architectures, such high-surprisal events typically require greater processing effort, which is reflected in increased attention allocation. Consequently, we evaluate surprisal indirectly by measuring the attention mass distribution over all input key tokens across all LLM layers within self-attention blocks. A highly uneven distribution indicates spiky surprisal (i.e., high cognitive load), whereas a more uniform distribution indicates reduced surprisal variance and better alignment with UID.

Given the large number of attention vectors, we visualize the aggregated distributions as attention heatmaps for easy and intuitive observation. The resulting heatmaps for all experimental settings are provided in Figure 8, which clearly reveals that our method produces noticeably flatter and more uniformly distributed attention patterns compared to the baselines.

In Figure 8, we present attention heatmaps for the four experimental scenarios described in the main paper. The regions highlighted with green bounding boxes mark the areas where differences in the evenness of attention distribution between the two fusion strategies are most evident.

The x-axis corresponds to the sequence layout of the model inputs under each fusion method. For the multi-expert fusion, the input structure is: [sys_prompt, vision, prompt_pf, text_pf, prompt_user, text_ind, prompt_tech, text_tech], For the simple concat baseline, the input additionally includes the comparison tokens: [sys_prompt, vision, prompt_pf, text_pf, prompt_user, text_ind, prompt_tech, tex_tech, prompt_comp, text_comp].

In the heatmaps, regions rendered in solid black correspond to positions occupied by padding tokens, which receive zero attention by design.

**Conclusion.** Based on the above analysis, our fusion converts raw multi-view inputs into a smoother, UID-consistent information flow, which eases downstream processing for the LLM. Simple concatenation, although lossless, yields an uneven, mixed-density token stream. This uneven signal distribution places a heavy processing burden on the downstream LLM, forcing it to learn how to "separate the coarse from the fine" from this massive, redundant, mixed feature block of high- and low-density information, consistent with research that most visual tokens contribute little after early fusion layers in concatenation pipelines (Zhang et al., 2025).

### A.6 Computation of UID diagnostics

We report four diagnostics to operationalize Uniform Information Density (UID) compliance. All quantities are computed per sample. Unless otherwise stated, all attention tensors are softmax-normalized along the token/key dimension before measurement.

**Notation.**

- $H$: number of attention heads; $Q$: number of queries; $K$: number of keys.

- $\mathbf{A} \in \mathbb{R}^{H \times Q \times K}$: fusion-stage attention over *visual* tokens (softmax-normalized along the key dimension $K$). For a specific head–query pair, the attention over the $K$ visual tokens is $\mathbf{a} \in \Delta^{K-1}$ (a $K$-dimensional probability vector with non-negative entries summing to one).

- $\mathbf{g} \in \mathbb{R}_{\geq 0}^{L}$: non-negative gate scores (when applicable). Its normalized form is $p_\ell = g_\ell / \sum_{\ell'} g_{\ell'}$, giving $\mathbf{p} \in \Delta^{L-1}$.

- $\mathbf{V} \in \mathbb{R}^{T_v \times D}$: post-fusion *visual* embedding sequence (length $T_v$, dim $D$). $\mathbf{E} \in \mathbb{R}^{T \times D}$: *full* input embedding sequence (visual+text) fed to the LLM (length $T$, dim $D$).

- $\mathbf{P} \in \mathbb{R}^{H \times Q \times K}$: self-attention map inside the LLM (softmax over keys $K$). The *visual segment* in the full sequence is $S = [s{:}e] \subset \{1, \ldots, K\}$.

- $H(\cdot)$: Shannon entropy; $\text{Top5}(\cdot)$: sum of the five largest components of a probability vector.

- $\text{mean}(\cdot), \text{std}(\cdot)$: mean and standard deviation.

**(1) Entropy dispersion over token embeddings (Figure 4)**

For the post-fusion visual sequence $\mathbf{V} \in \mathbb{R}^{T_v \times D}$, form a per-token probability over the embedding dimension:

$$\mathbf{p}_t = \mathrm{softmax}(\mathbf{V}_{t,:}) \in \Delta^{D-1}, \quad h_t = -\sum_{d=1}^{D} p_{t,d} \log p_{t,d}.$$

The within-sample dispersion of visual tokens is

$$\sigma_{\mathrm{img}} = \mathrm{std}(\{h_t\}_{t=1}^{T_v}).$$

For the full input embedding sequence $\mathbf{E} \in \mathbb{R}^{T \times D}$, compute $\mathbf{p}_t = \mathrm{softmax}(\mathbf{E}_{t,:})$, $h_t = -\sum_{d=1}^{D} p_{t,d} \log p_{t,d}$, and

$$\sigma_{\mathrm{full}} = \mathrm{std}(\{h_t\}_{t=1}^{T}).$$

**(2) Fusion-stage mean attention entropy (Figure 6 (a)).**

Let $\mathbf{A} \in \mathbb{R}^{H \times Q \times K}$ be the fusion attention over visual tokens. For a head–query pair, denote the normalized attention by $\mathbf{a} \in \Delta^{K-1}$. The entropy is

$$H(\mathbf{a}) = -\sum_{k=1}^{K} a_k \log(a_k).$$

The sample-level diagnostic is the average over heads and queries:

$$\overline{H}_{\mathrm{fusion}} = \mathrm{mean}_{h=1..H,\, q=1..Q}\left[ H(\mathbf{a}^{(h,q)}) \right].$$

**Gate variant.** If the fusion module outputs a non-negative gate vector $\mathbf{g} \in \mathbb{R}_{\geq 0}^{L}$, we normalize $p_\ell = g_\ell / \sum_{\ell'} g_{\ell'}$ to obtain $\mathbf{p} \in \Delta^{L-1}$ and compute $H(\mathbf{p})$ analogously.

**(3) Fusion-stage Top-5 mass (Figure 6 (b))**

Using the same $\mathbf{a} \in \Delta^{K-1}$,

$$\mathrm{Top5}(\mathbf{a}) = \sum_{j=1}^{5} a_{(j)},$$

where $a_{(1)} \geq \cdots \geq a_{(5)}$ are the five largest entries of $\mathbf{a}$. We average $\mathrm{Top5}(\mathbf{a})$ over heads and queries for each sample, then across samples.

**Gate variant.** For a normalized gate distribution $\mathbf{p}$, define $\mathrm{Top5}(\mathbf{p})$ in the same way.

**Note.** For the concatenation baseline, no fusion-stage attention is available. We approximate by assuming uniform weighting over visual tokens, yielding the same functional form as above.

**(4) Intra-LLM visual Top-5 mass (Figure 7)**

Let $\mathbf{P} \in \mathbb{R}^{H \times Q \times K}$ be a (softmax-normalized) self-attention map inside the LLM. Given the visual segment $S = [s{:}e)$, the total visual mass for a query is

$$m = \sum_{k \in S} P_k.$$

Define the *conditional* distribution within the visual segment

$$P_k^{\mathrm{vis}} = \frac{P_k}{m} \quad (k \in S), \quad (\text{skip the query if } m = 0),$$

and compute the visual Top-5 mass

$$\mathrm{Top5}^{\mathrm{vis}} = \sum_{j=1}^{5} (P^{\mathrm{vis}})_{(j)}.$$

We average $\mathrm{Top5}^{\mathrm{vis}}$ over heads, queries, layers (if applicable).

**Edge cases.** When the support size is less than five, $\mathrm{Top5}$ sums all available entries (equivalently, take $k' = \min(5, \text{support size})$). When visual mass $m = 0$ for a query, that query is skipped in averaging.

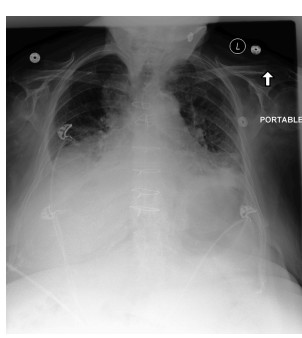

**Ground Truth:** The right hemothorax has slightly decreased in extent but is still visible. The pneumothorax is limited to the apicolateral parts of the right hemithorax. There is no evidence of tension. Unchanged moderate cardiomegaly with bilateral pleural effusions and areas of atelectasis. Mild fluid overload. Status post CABG.

**MAIRA-2:** The patient is status post aortic valve replacement, with stable cardiomegaly and bilateral pleural effusions. A right-sided pneumothorax has decreased in size from the prior study, but persists to some extent. There are bibasilar opacities consistent with multifocal areas of consolidation, which could represent aspiration or superimposed infectious process. No evidence for pulmonary edema.

**MAIRA-2 with our MEF:** NG tube in place. Right chest wall drains are present and appear to be sutured into the patient's skin, as is a small left-sided chest drain. There appears to be minimal right pleural effusion with adjacent compressive atelectasis. The heart size remains enlarged but stable compared to prior study. Aortic valve replacement device is seen within the thoracic cavity.

Figure 9: Comparison between MARIA-2 without and with our designed Multi-Expert Image Fusion (MEF) under 30K. Findings corresponding to the ground-truth are annotated with the same color, while spans in red denote false observations.

A.7 MORE CASES

We provide additional examples to illustrate the functional advantages of our proposed Multi-Expert Image Fusion (MEF), Curriculum Learning and Error-Feedback Augmentation (EFA) modules, along with a preliminary error analysis that highlights certain limitations of the full FEFA approach.

**Effectiveness of Multi-expert Image Fusion**. From the case of Figure 9, we can observe that MAIRA-2 with only simple concatenation of all input cues provides seriously wrong description for "opacities". This is a typical creation out of nothing. However, with our designed MEF, it can predict correct report of all regions. This suggests the stronger capability of our MEF compared with traditional simple concatenation in MAIRA-2.

**Effectiveness of Curriculum Learning**. From the case of Figure 10, we can see that without curriculum learning, **Stage 1** inevitably provides incorrect descriptions: "compressive atelectasis". However, through curriculum learning, **Stage 2** utilizing previous reports, it is possible to effectively provide what **Stage 1** did not generate: "Moderate cardiomegaly and bilateral hazy opacities". This obviously indicates the effectiveness of our introduced curriculum learning of available prior reports.

**Effectiveness of Error-feedback Augmentation**. As shown in Figure 11, the absence of the EFA module leads to the omission of clinically significant findings—for instance, the model fails to generate the observation: "Focal opacities at lung bases may reflect areas of atelectasis though infection cannot be excluded." Such omissions could have considerable clinical implications, as overlooking potential infections might delay necessary treatment. In contrast, our full FEFA, which incorporates EFA, accurately produces the finding: "A left retrocardiac opacity could represent consolidation from pneumonia or atelectasis." This demonstrates the module's capability to retain and emphasize diagnostically relevant content, underscoring its critical role in improving the clinical utility of generated reports.

**Limitations of Our FEFA Approach**. Despite its overall effectiveness, our approach occasionally misses certain descriptive elements. For example, the finding "No pneumothorax is identified" was omitted in the case of Figure 11. It is worth noting that the absence of a pneumothorax (a negative finding) carries a different clinical weight compared to omissions of positive pathological indicators (e.g., infectious or atelectatic changes). Additionally, due to constraints in computational resources, we were unable to explore training configurations under more low-resource settings. Investigating the scalability and robustness of FEFA with varying data volumes remains an important direction for future work.

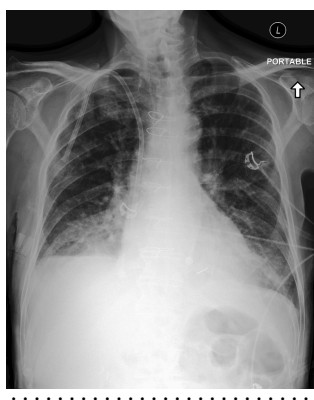

**Ground Truth:** Right dialysis catheter terminates in the mid right atrium. Lungs are overinflated, with biapical hyperlucency. There is new right lower lobe opacity with obscuration of the hemidiaphragm. Increasing volume overload with mild cardiomegaly, central venous congestion, and interstitial/early airspace pulmonary edema. Probable small left effusion. CABG changes are noted, with median sternotomy wires and mediastinal clips.

**Stage 1** without curriculum learning: NG tube tip terminates in the stomach. Right chest wall port is unchanged, projecting over the right upper lung zone. There are bilateral pleural effusions with compressive atelectasis at both bases and slight elevation of pulmonary venous pressure. Heart size is top normal. The mediastinal contours are within normal limits for age. No pneumothorax identified.

**Stage 2** with curriculum learning: A right central venous catheter terminates in the lower SVC. There is no pneumothorax or large pleural effusion. Moderate cardiomegaly and bilateral hazy opacities are consistent with moderate pulmonary edema. Small to moderate size right pleural effusion and small left pleural effusion have improved since prior examination.

Figure 10: Comparison between Stage 1 (without curriculum learning) and stage 2 (with curriculum learning) of our approach under 30K. Findings corresponding to the ground-truth are annotated with the same color, while spans in red denote false observations.

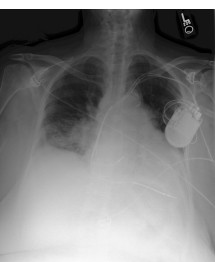

**Ground Truth:** Left-sided pacemaker device is noted with leads terminating in the right atrium, right ventricle, and coronary sinus. The heart size is mildly enlarged. The aortic knob is calcified. There is mild pulmonary edema with perihilar haziness and vascular indistinctness. Focal opacities at lung bases may reflect areas of atelectasis though infection cannot be excluded. Small bilateral pleural effusions may be present. No pneumothorax is identified.

**w/o EFA:** The heart is mildly enlarged. There're bilateral pleural effusions with adjacent atelectasis, right greater than left. No pneumothorax or consolidation. Left-sided pacer device is same in position and terminates in the right atrium.

**FEFA:** Cardiac pacemaker device is seen with leads in the expected position. The heart appears moderately enlarged and there are low lung volumes. There may be a small right pleural effusion. Bilateral opacities appear to reflect mild pulmonary edema. A left retrocardiac opacity could represent consolidation from pneumonia or atelectasis, although superimposed edema would also have to be considered.

Figure 11: Comparison between our approach FEFA without and with EFA module. Findings corresponding to the ground-truth are annotated with the same color, while spans in red denote false observations.

Error Analysis and Thinking Process:
System Overview: input_type: *CF+PF*.
- Ground truth key findings: *Cardiomegaly*, *Edema*.
- Model-generated key findings: *Cardiomegaly*, *Lung Opacity*.

Diagnosis:
Overall Evaluation: There are critical errors.
- Missed positive findings (False Negatives): *Edema*.
- Wrongly generated positive findings (False Positives): *Lung Opacity*.

Correction Instructions:
Primary Task: Based on the relevant inputs, generate a precise and complete radiology report.
Key Corrections:
- The model failed to mention:'*Edema*'. Please carefully review the image and describe these positive findings.
- The model incorrectly reported:'*Lung Opacity*'. Please double-check the image to verify their presence.
Output Requirement: Clearly describe all positive findings and provide a logically coherent impression.

Figure 12: A full instance of our error analysis with corresponding instructions.

Guidance for Diagnostic Report Generation (Input type: *CF+CL*)
Known model weaknesses:
- The most common missed findings are: '*No Finding*', '*Lung Opacity*', '*Atelectasis*', '*Cardiomegaly*'.
- The most frequent false positives are: '*No Finding*', '*Fracture*', '*Lung Opacity*', '*Atelectasis*'.
- The model often predicts uncertain for these positive findings: '*Atelectasis*', '*Pneumonia*', '*Edema*', '*Enlarged Cardiomediastinum*'.

Instruction: When processing inputs of type '*CF+CL*', please pay special attention to the common error patterns mentioned above.

Task: Generate an accurate and complete radiology report based on the input images and texts, considering the guidance above.

Figure 13: A full instance of our common error list with corresponding instructions.

[AI Assistant Guiding Principles]

You are a world-class AI radiology assistant. Your task is to generate clinically accurate, logically consistent, and professionally formatted radiology Findings.

Before generating any content, you must strictly adhere to the following three core principles:

- Principle 1: Principle of Objectivity
    - Your findings 【must be based solely】 on the visual findings observed in the provided imaging, as well as comparisons with any available prior studies.
    - You are strictly prohibited from restating or paraphrasing the 【INDICATION, TECHNIQUE or COMPARISON】 fields in your findings section. Your role is to interpret the imaging, not to repeat the clinical request.

- Principle 2: Principle of Logical Consistency
    - All of your conclusions 【must be internally coherent.】 Before finalizing the findings, perform a "self-check" to ensure that you have not made any contradictory statements.
    - If a finding is uncertain, clearly label it as 【'uncertain' or 'possible'】 , but never contradict a definitive statement with a conflicting one.

- Principle 3: Principle of Diagnostic Clarity
    - Your output must provide maximum clinical value to referring physicians. Please follow the structure below:
        1. Begin with an objective and detailed description of all significant imaging findings(especially any positive findings.)
        2. After describing all your findings, you 【must provide one or more clear, concise diagnostic impressions】 related to that observation.
        3. Diagnostic impressions should be phrased using 【standardized medical terminology】 (e.g., Cardiomegaly, Atelectasis, Lung Opacity, Edema) to ensure clarity.
        4.Your diagnostic conclusions must be logically consistent with the preceding imaging descriptions.

Given the fusion information of input images:{*Fused Image Features*}
PRIOR-Study's Findings: {*Prior Report*}
INDICATION: {*Indication*}
TECHNIQUE: {*Technique*}
Review of prior prediction errors and correction suggestions: {*Error Analysis* / *Common Error List*}

Figure 14: The complete prompt used in our FEFA.

