# OpenReview forum: "FUSION-BASED ERROR-FEEDBACK AUGMENTATION FOR LOW-RESOURCE RADIOLOGY REPORT GENERATION"
_ICLR.cc/2026/Conference — Submitted to ICLR 2026_

### Official Review · Reviewer_11TT · 2025-10-28

**Soundness:** 3
**Presentation:** 2
**Contribution:** 2
**Rating:** 6
**Confidence:** 2

**Summary:**

This paper proposes FEFA (Fusion-based Error-Feedback Augmentation), a novel approach for low-resource radiology report generation (RRG). It introduces two core mechanisms: (1) Multi-expert Image Fusion Module: Using self-attention, cross-attention, and gating mechanisms to dynamically fuse multi-view input images (frontal, lateral, prior); (2) Error-feedback Augmentation: Incorporating sample-level error analysis and common error lists generated by a large language model (LLM) to enable self-correction during training. Experiments on MIMIC-CXR show that FEFA achieves 94% of the clinical efficacy score of the best existing method with only 8% of the supervised data, outperforming all other competitors. The work demonstrates significant improvements in data efficiency and model adaptability for real-world clinical scenarios.

**Strengths:**

1.	The combination of multi-expert fusion and error-feedback augmentation provides both technical and practical benefits.
2.	Experimental results demonstrate the effectiveness of FEFA with limited data and outperform other baselines.
3.	The paper provides a theoretical framework with the UID theory, even though it requires further quantification.
4.	The approach is highly relevant for low-resource medical AI, which is an important and under-explored area.

**Weaknesses:**

1.	The error-feedback augmentation method needs more in-depth analysis to show its robustness in noisy environments or when dealing with imbalanced error feedback.
2.	The impact of curriculum learning could be more clearly defined and validated independently from other components.
3.	The UID theory is an interesting hypothesis but requires statistical validation to substantiate its role in improving the model's interpretability and performance.

**Questions:**

1.	Can the error-feedback mechanism handle noisy or inconsistent error feedback, and how does it affect the model's performance in such cases?
2.	How does curriculum learning contribute to the model's improvement in low-resource settings? Are the results independent of the other modules?
3.	Would the error-feedback strategy still be effective if the model is trained with a larger dataset? How would it scale?
4.	Can you provide more quantitative analysis or statistical validation for the UID theory claims?
5.	How would FEFA perform when applied to datasets with more heterogeneous, real-world data or in cross-domain applications?

---

> ### Author Response · Authors · 2025-12-02
> **Response to Reviewer 11TT (Page 1/4)**
>
> Thank you for your valuable feedback to help us improve our paper. We have revised our paper based on your feedback. We detail our response below and please kindly let us know if our response addresses your concerns.
>
> >**W1:** The error-feedback augmentation method needs more in-depth analysis to show its robustness in noisy environments or when dealing with imbalanced error feedback.
> >
> >**Q1:** Can the error-feedback mechanism handle noisy or inconsistent error feedback, and how does it affect the model's performance in such cases?
>
> **A for W1 & Q1:**
> We thank you for this critical question regarding the reliability of the error-feedback mechanism.
>
> To quantitatively evaluate the error-feedback mechanism's robustness in noisy environments, **we performed a controlled sensitivity study where we systematically injected noise into the feedback**. Specifically, we **constructed three tiers of feedback quality based on the mismatch frequency** between second-stage predictions and ground truth:
>  - Tier 1 (high-quality, default): Prompting with the Top-4 high-frequency errors (real "pain points").
>  - Tier 2 (mild noise): Prompting with 4 medium-frequency errors (suboptimal/less relevant feedback).
>  - Tier 3 (severe noise): Prompting with the bottom-4 low-frequency errors (simulating noisy, inconsistent, or hallucinated feedback signals).
>
> All experiments were conducted using the 30k model, evaluating the 14 CheXbert labels (treating uncertain labels as negative). **As shown in Table R1, the results reveal a clear performance gradient**.
>
> - The results show the error-feedback mechanism demonstrates robustness, as the overall accuracy improves even under mild and severe noise (Tier 2 & 3). This suggests that the process of **error-aware SFT itself encourages better representation learning, regardless of the prompt's perfection**.
>
> - The "unmentioned labels" metric is particularly insightful. **In Tier 2 (Mild Noise) and Tier 3 (Severe Noise), the model successfully improved the specific errors it was told to fix** (even if they were rare), **but at the cost of degrading other unmentioned labels**. This "over-correction" phenomenon confirms that the model is actively attending to and semantically following the feedback instructions.
>
> Overall, the mechanism is robust to mild inconsistencies, showing consistent gains even with imperfect feedback. However, **under severe noise, the model's targeted improvements come with measurable trade-offs, reinforcing the importance of our frequency-based feedback selection strategy**, which keeps the feedback aligned with the model's actual systematic errors.
>
> We have revised the paper and put these details in our **new Section 4.4**.
>
> **Table R1:** Performance gains of FEFA (trained with 30k samples) across each clinical observation in CX-14 when using errors of different frequencies as feedback during inference. "Obs." is short for observations and positive $\Delta$Acc. are highlighted in bold.
>
> |14 Clinical Observations Labeled by CheXbert|FEFA w/o Error-aware Prediction|FEFA (w/ Most Frequent Errors)||FEFA w/ Moderate Frequent Errors||FEFA w/ Least Frequent Errors||
> |-|-|-|-|-|-|-|-|
> ||Accuracy|$\Delta$Acc.|Obs. in Feedback|$\Delta$Acc.|Obs. in Feedback|$\Delta$Acc.|Obs. in Feedback|
> |Enlarged Cardiomediastinum|87.85|**0.16%**|✓|-1.99%|✓|-1.54%||
> |Cardiomegaly|70.58|-2.03%|✓|-1.95%|✓|-2.28%||
> |Lung Opacity|62.82|**1.14%**|✓|**0.24%**||**0.20%**|✓|
> |Lung Lesion|92.32|**0.93%**||**1.06%**|✓|**0.53%**|✓|
> |Edema|79.20|**1.30%**|✓|**1.18%**|✓|**0.08%**|✓|
> |Consolidation|92.32|**1.63%**||-0.08%|✓|**0.16%**|✓|
> |Pneumonia|93.25|**1.46%**|✓|**1.38%**|✓|**0.61%**|✓|
> |Atelectasis|69.08|**1.22%**|✓|-1.95%||**1.02%**||
> |Pneumothorax|96.46|**1.46%**||**0.81%**|✓|**1.02%**|✓|
> |Pleural Effusion|76.68|**1.58%**|✓|**0.49%**|✓|-0.45%||
> |Pleural Other|95.73|**0.65%**||**0.37%**|✓|-0.20%|✓|
> |Fracture|94.80|-0.41%||-0.73%|✓|-1.10%|✓|
> |Support Devices|84.19|-0.08%||-1.26%|✓|-0.65%||
> |No Finding|88.58|**1.02%**|✓|**3.01%**|✓|**2.93%**||
> |**Avg. of All Obs.**|*84.56*|*0.72*%||*0.04%*||*0.02%*||
> |**Avg. of Obs. in Feedback**|-|*0.73%*||*0.19%*||*0.16%*||
> |**Avg. of Obs. Not in Feedback**|-|*0.70%*||*-0.85%*||*-0.16%*||

---

> ### Author Response · Authors · 2025-12-02
> **Response to Reviewer 11TT (Page 2/4)**
>
> >**W2:** The impact of curriculum learning could be more clearly defined and validated independently from other components.
> >
> >**Q2:** How does curriculum learning contribute to the model's improvement in low-resource settings? Are the results independent of the other modules?
>
> **A for W2 & Q2:**
> We appreciate your request to isolate the contribution of the curriculum learning (CL) strategy.
>
> To prove that CL provides benefits independent of our proposed architectural changes, we conducted a specific ablation on the 5k low-resource split. As shown in Table R2, there are four sets of comparisons:
>  - Standard baseline (MAIRA-2 without any module we proposed) vs. baseline + CL.
>  - Multi-expert Image Fusion (MEF) vs. MEF + CL.
>  - Error-feedback Augmentation (EFA) vs. EFA + CL
>  - MEF + EFA vs. MEF + EFA + CL (i.e., our FEFA)
>
> **Across all four comparisons, curriculum learning consistently improves performance, most notably on the clinically oriented metrics**, confirming that its effects are independent, complementary, and particularly impactful under low-resource constraints.
>
> We have revised the paper and rearranged the results in Table R2 as additional ablation studies in **Appendix A.3**.
>
> **Table R2:** Thorough ablation experiments towards curriculum learning on MIMIC-CXR. All models are trained with 5k samples. "FEFA w/o EFA & CL" and "FEFA w/o EFA" have also been shown in Table 4 (i.e., variants (a) and (b)).
>
> |Model|Multi-expert Image Fusion|Curriculum Learning|Error-feedback Augmentation|B-4|MTR|R-L|CX-14 (u$\to$p) Mac-F1|CX-14 (u$\to$p) Mic-F1|CX-14 (u$\to$n) Mac-F1|CX-14 (u$\to$n) Mic-F1|
> |-|-|-|-|-|-|-|-|-|-|-|
> |MAIRA-2||||4.67|23.44|20.28|25.96|36.99|22.27|35.98|
> |MAIRA-2 w/ CL||✓||**5.43**|**25.70**|**20.72**|**27.44**|**38.47**|**22.78**|**37.35**|
> ||||||||||||
> |**FEFA** w/o EFA & CL|✓|||**4.42**|25.32|19.53|28.27|40.04|23.73|39.89|
> |**FEFA** w/o EFA|✓|✓||4.31|**26.21**|**19.84**|**31.99**|**44.62**|**27.40**|**44.43**|
> ||||||||||||
> |**FEFA** w/o MEF & CL|||✓|4.84|26.35|20.06|26.64|37.49|22.70|36.72|
> |**FEFA** w/o MEF||✓|✓|**5.34**|**26.56**|**20.49**|**28.28**|**39.46**|**22.88**|**38.00**|
> ||||||||||||
> |**FEFA** w/o CL|✓||✓|**5.79**|26.44|**21.26**|30.54|40.15|26.13|38.77|
> |**FEFA**|✓|✓|✓|5.64|**27.34**|20.93|**35.20**|**47.59**|**30.36**|**47.27**|

---

> ### Author Response · Authors · 2025-12-02
> **Response to Reviewer 11TT (Page 3/4)**
>
> >**W3:** The UID theory is an interesting hypothesis but requires statistical validation to substantiate its role in improving the model's interpretability and performance.
> >
> >**Q4:** Can you provide more quantitative analysis or statistical validation for the UID theory claims?
>
> **A for W3 & Q4:**
> We appreciate your interest in the UID hypothesis and the request for stronger statistical validation. However, we need to clarify one fact: **the UID theory does not contribute to our performance improvement; we only use its content to explain why performance is improved.** And in the main paper (Table 4, Figure 4) and Appendix A.5 (Figures 6 and 7), **we have already provided statistical evidence supporting our UID-based interpretation**, and used these analyses together with UID theory explained the performance advantages of our model.
>
> In the revision, we provide **a new quantitative analysis** based on Surprisal as below, which more rigorously supports our UID-related claims.
>
> In our work, we conceptualize the Vision-LLM interaction as a communication channel, where **the fusion module serves as the "Speaker"** and **the LLM as the "Listener".** According to UID theory, an optimal speaker should deliver information with uniform density, reducing the listener's processing burden. In information theory terms, **this burden is quantified by Surprisal**.
>
> Conventional surprisal estimation based on perplexity is not applicable here, because we are **measuring the surprisal of the information received by the LLM** and this is an intermediate state, lacking ground truth to calculate PPL. Therefore, we start from the definition of surprisal. According to the definition of surprisal, low probability and unpredictable events carry more information and therefore have higher surprisal [1]. **This also indicates that the model needs to pay more attention to this**. So we compute the attention mass distribution across all layers of the LLM for each input key token within the self-attention blocks. **A highly uneven distribution indicates spiky surprisal (i.e., high cognitive load), whereas a more uniform distribution indicates reduced surprisal variance and better alignment with UID**. Due to the large number of data, we plotted it into an attention heatmap for easy and intuitive observation, which is added to the **Appendix A.5**.
>
> Across the four experimental scenarios presented in the paper, the heatmaps consistently show that our multi-expert fusion produces markedly **more uniform attention patterns** over the input sequence compared to the baseline. This quantitative evidence demonstrates that our method reduces surprisal variance of the information input to the LLM, providing **quantitative support that the fused visual representation more closely adheres to UID principles and improves the interpretability and stability of the model's downstream reasoning**.
>
> **Reference**
>
> [1] Clark, Thomas Hikaru, et al. "A cross-linguistic pressure for Uniform Information Density in word order." *Transactions of the Association for Computational Linguistics* 11 (2023): 1048-1065.
>
> ---
>
> >**Q3:** Would the error-feedback strategy still be effective if the model is trained with a larger dataset? How would it scale?
>
> **A for Q3:**
> To evaluate whether the error-feedback strategy remains effective at larger data scales, **we conducted an additional controlled experiment by increasing the training data from 30k to 60k samples**.
>
> As shown in Table R3, we confirm that the error-feedback strategy remains highly effective. The error-feedback stage continues to provide consistent and meaningful improvements even at this larger scale. This demonstrates that **the proposed feedback mechanism is not limited to low-resource settings, but remains beneficial as the data scale grows**.
>
> **Table R3:** Ablated performance of FEFA on MIMIC-CXR trained with 60k samples.
>
> |Model|B-4|MTR|R-L|CX-14 (u$\to$p) Mac-F1|CX-14 (u$\to$p) Mic-F1|CX-14 (u$\to$n) Mac-F1|CX-14 (u$\to$n) Mic-F1|
> |-|-|-|-|-|-|-|-|
> |**FEFA** w/o Error-Feedback Augmentation|6.79|28.79|23.07|38.85|52.9|34.54|53.39|
> |**FEFA**|**10.13**|**31.69**|**28.02**|**41.71**|**53.88**|**37.25**|**53.55**|

---

> ### Author Response · Authors · 2025-12-02
> **Response to Reviewer 11TT (Page 4/4)**
>
> >**Q5:** How would FEFA perform when applied to datasets with more heterogeneous, real-world data or in cross-domain applications?
>
> **A for Q5:**
> Thank you for considering the practical deployment of the model. We argue that FEFA is specifically engineered to handle the heterogeneity inherent in real-world clinical environments.
>
> First, it is important to clarify that **MIMIC-CXR is inherently a "real-world" dataset**, sourced from the routine workflow of a busy medical center (BIDMC). It is not a curated "toy" dataset as it **features significant heterogeneity** primarily in two dimensions:
>
> - **Diverse real imaging artifacts and acquisition noise in films**, such as support devices, tubes, and patient motion (common in ICU settings).
> - **Uncertain number of views and accessibility of longitudinal data**, as the report could be written based on solely one image, or multiple frontal and lateral films captured at different times.
>
> The strong performance of our model on this dataset attests to its robustness against real-world imaging noise and intricate connections between multiple views and modalities.
>
> **To explicitly address the "cross-domain" capability, we evaluated our method on the IU X-ray dataset** (sourced from a completely different institution, Indiana University, with different patient demographics and reporting styles). As shown in Table R4, **our method maintains consistent performance with the main paper on this dataset**, which confirms that our architecture captures generalized radiological features rather than overfitting to the specific distribution or linguistic style of a single hospital (MIMIC-CXR).
>
> **Table R4:** Performance comparisons with SOTA MAIRA-2 on varing low-resource training sets on IU X-ray.
>
> |Training Scale|Model|B-4|MTR|R-L|CX-14 (u$\to$p) Mac-F1|CX-14 (u$\to$p) Mic-F1|CX-14 (u$\to$n) Mac-F1|CX-14 (u$\to$n) Mic-F1|
> |-|-|-|-|-|-|-|-|-|
> |5k|MAIRA-2|6.06|32.33|23.33|9.42|23.88|8.84|25.43|
> ||**FEFA**|**7.62**|**34.64**|**24.92**|**19.83**|**46.20**|**18.83**|**48.72**|
> |15k|MAIRA-2|6.50|32.45|23.93|13.67|35.45|12.48|38.15|
> ||**FEFA**|**7.33**|**33.87**|**25.27**|**22.66**|**46.77**|**20.75**|**49.41**|
> |30k|MAIRA-2|6.74|32.75|24.47|21.27|35.09|18.88|37.49|
> ||**FEFA**|**7.22**|**34.79**|**25.93**|**27.15**|**46.76**|**25.82**|**49.48**|
> |400k+ (Official)|MAIRA-2|8.29|33.67|28.30|31.29|50.44|29.34|52.56|

---

### Official Review · Reviewer_fkaR · 2025-10-29

**Soundness:** 3
**Presentation:** 2
**Contribution:** 3
**Rating:** 4
**Confidence:** 4

**Summary:**

This manuscript proposes FEFA, a multi-expert image fusion module that additionally incorporates error-feedback augmentation, with the augmentation strategy utilizing a large language model (LLM). FEFA compares favourably against the state-of-the-art MAIRA-2, using only 8% of the supervised data. In particular, a multi-expert fusion module that addresses MAIRA-2's issues with fusion by concatenation, is proposed. However, major details especially as pertaining to the LLM are missing.

**Strengths:**

- (Originality) Proposal of multi-expert image fusion module, comprising three separate experts to handle various combinations of radiological views

 - (Quality) Detailed justification, description and reporting of Uniform Information Density metrics for quantifying understandability and supporting the use of attention-based fusion

 - (Clarity) Provided case studies in the Appendix

 - (Significance) Empirical finding on possible detrimental impact of lateral radiological image views, without prior frontal images

**Weaknesses:**

- Severe lack of details about the downstream LLM (even its architecture; it is just referred to as "an LLM" in the Appendix), especially regarding parameter settings and LLM fine-tuning in the curriculum learning and error-aware SFT stages (Figure 3); only broad objective formulae (Equations 6-8) are provided

 - Lack of direct comparison for FEFA against competing SOTA methods on the same training datasets (Table 2), with no attempt to minimally match training scales, at least for the lowest scale (40k)

 - Again, while the claim of achieving 94% of MAIRA-2 performance using only 8% of the supervised data appears impressive, the tradeoff for most of the competing models (as from Table 2) is not presented, for the direct comparison on low-resource datasets (Table 3). As such, the true contribution of FEFA for low-resource modelling could be further justified

 - Section 4.3 might be edited for clarity, as it is not immediately clear as to what ablations (a), (b) and (c) represent, or that FEFA is essentially equivalent to MAIRA-2 + (a), (b), (c) and also test-time feedback; since the ablations appear additive, combinations such as error-feedback augmentation without curriculum learning also do not appear to be explored

**Questions:**

1. In Section 3.3, for multi-expert image fusion, it is stated that fused image features are output, if only current images (CF or CF+CL) are available. However, it is then stated that the output is a fixed-length representation. It might be clarified as to whether the Self-Att(.) and Cross-Att(.) dimensionalities are designed to be equal in size, and how this output size was determined.

2. Related to the above, in Section 4.4, it is then stated that the fused representation contains no more raw information than concatenation, which appears to imply that the size of the fused representation is equal to the maximum size of concatenated views (three views). However, it also appears that the objective should be to maximize FEFA performance (on the NLG and CX-14 metrics), and not to constrain input dimensionality. If so, it could be discussed as to whether a larger fused representation might be appropriate.

3. Given the observation on lateral views, it might be clarified as to whether these lateral views are used for clinical diagnosis in current medical practice, and if so, what conditions rely more on lateral views for diagnosis.

---

> ### Author Response · Authors · 2025-12-02
> **Response to Reviewer fkaR (Page 1/4)**
>
> Thank you for your constructive comments and suggestions. We have revised our paper according to your comments. We respond to your questions below and would appreciate it if you could let us know if our response  addresses your concerns.
>
> >**W1:** Severe lack of details about the downstream LLM (even its architecture; it is just referred to as "an LLM" in the Appendix), especially regarding parameter settings and LLM fine-tuning in the curriculum learning and error-aware SFT stages (Figure 3); only broad objective formulae (Equations 6-8) are provided
>
> **A for W1:**
> We appreciate your comment and apologize for the insufficient description of the downstream LLM and its fine-tuning strategy. We have substantially clarified these details in the revised manuscript.
>
> As described in Appendix A.1 (but admittedly not emphasized clearly), **the "LLM" used in our main experiments is Vicuna-7B-v1.5**, a standard decoder-only autoregressive Transformer. We selected this backbone to ensure a fair comparison with MAIRA-2, which uses the same model family.
>
> Details of fine-tuning strategies across stages (Curriculum Learning & Error-Aware SFT) are shown below:
>
> - Modules trained during Curriculum Learning: **We fine-tune the LLM using LoRA (rank = 8, α = 32) with trainable modules {q_proj, k_proj, v_proj}**. In addition, both **the proposed multi-expert fusion module and the visual projector remain fully trainable throughout all curriculum stages** to preserve proper alignment between visual features and the LLM token space.
> - Modules training in Stage Transition: The error-aware SFT stage initializes from the curriculum-learning checkpoint. **We continue training the same set of modules (LoRA adapters, projector, and fusion module) using the augmented error-feedback data**. We have **updated the caption of Figure 3** to explicitly indicate that the LLM is trained jointly with multi-expert fusion module and the adapter in all stages.
>
>
> Additionally, all training hyperparameters, including optimizer settings, learning-rate schedule, batch size, number of epochs, and warmup strategy are summarized in **Appendix A.1**. In the revision, we have reorganize this section to make these details more visible and easier to interpret.

---

> ### Author Response · Authors · 2025-12-02
> **Response to Reviewer fkaR (Page 2/4)**
>
> >**W2:** Lack of direct comparison for FEFA against competing SOTA methods on the same training datasets (Table 2), with no attempt to minimally match training scales, at least for the lowest scale (40k)
>
> **A for W2:**
>
> Thank you for highlighting the concern regarding the mismatch between our training scale and the 40k scale used by competing methods in Table 2.
>
> Although our experiments did not exactly reproduce the 40k setting, we contend that the comparison remains fair.  **In fact, it presents a more conservative benchmarking setup for our method**. All our subsets are obtained by random sampling from the same parent set as the 40k data set, and **it is widely observed in large vision-language training that performance correlates positively with dataset size. Therefore, by comparing FEFA trained on {5k,15k,30k} against competing approaches trained on 40k, we established a stricter evaluation setting for ourselves**.
>
> **Despite this stricter evaluation, FEFA trained on 30k still outperforms all competing methods**, demonstrating its strong data efficiency in low-resource medical image-text learning.
>
> To further address the concern that our performance might not hold at larger scales (e.g., 40k or beyond), we conducted an additional controlled experiment where the training data was increased from 30k to 60k. As shown in Table R1, the model exhibits a clear monotonic improvement from 30k → 60k. At 60k, the model reaches 96-98% of the full-data SOTA performance, confirming that the model continues to benefit from additional supervision without saturation.
>
> **Given this monotonic scaling trend, it is highly plausible that FEFA trained on a 40k split would perform at least as well as, and likely better than, the reported 30k results**. Therefore, the performance advantage over competing SOTA models would remain intact or potentially widen when evaluated at the same training scale.
>
> Therefore, while the scales are not identical, the comparison still provides strong and conservative evidence of FEFA's superior data efficiency.
>
> **Table R1**: Performance of FEFA on MIMIC-CXR trained with 60k samples. CX-5 represents the classification performance for five specific diseases (Atelectasis, Cardiomegaly, Consolidation, Edema, and Pleural Effusion) within CX-14. "%" indicates the ratio relative to the performance of official MAIRA-2. Values above 95% are highlighted in bold.
>
> |Training Scale|Model|B-4|MTR|R-L|CX-14 (u$\to$p) Mac-F1|CX-14 (u$\to$p) Mic-F1|CX-14 (u$\to$n) Mac-F1|CX-14 (u$\to$n) Mic-F1|CX-5 (u$\to$p) Mac-F1|CX-5 (u$\to$p) Mic-F1|CX-5 (u$\to$n) Mac-F1|CX-5 (u$\to$n) Mic-F1|
> |-|-|-|-|-|-|-|-|-|-|-|-|-|
> |30k|**FEFA**|6.84|30.66|22.69|38.93|52.24|34.51|53.18|47.98|54.83|42.96|52.72|
> ||%|49.2%|87.8%|70.8%|90.4%|93.8%|86.6%|94.8%|93.5%|93.3%|91.5%|93.5%|
> |60k|**FEFA**|10.13|31.69|28.02|41.71|53.88|37.25|53.55|50.62|57.31|45.43|54.15|
> ||%|72.9%|90.7%|87.4%|**96.8%**|**96.7%**|93.5%|**95.5%**|**98.6%**|**97.5%**|**96.8%**|**96.1%**|
> |400k+ (Official)|MAIRA-2|13.89|34.93|32.07|43.07|55.70|39.86|56.10|51.34|58.77|46.93|56.37|
>
> ---
>
> >**W3:** Again, while the claim of achieving 94% of MAIRA-2 performance using only 8% of the supervised data appears impressive, the tradeoff for most of the competing models (as from Table 2) is not presented, for the direct comparison on low-resource datasets (Table 3). As such, the true contribution of FEFA for low-resource modelling could be further justified
>
> **A for W3:**
> We appreciate your thoughtful comment regarding the comparison in Table 3. Your concern is well taken: while Table 3 focuses on low-resource baselines, we did not retrain the full-scale SOTA models from Table 2 under the same low-resource setting.
>
> The reason is twofold. First, **it is a well-established empirical observation in deep learning that model performance consistently degrades as the amount of training data decreases**. Therefore, for any Competitor C,  its performance when trained on low-resource datasets would be **worse, not better** than its full-data result.
>
> Second, **as shown in Table 2 and Table 3, our method trained on just 30k data already outperforms competing models trained on the 160k dataset**. Showing that FEFA surpasses the full-data performance of these models while using only a small fraction of the data. **This provides a far stronger justification of FEFA's contribution than simply outperforming their artificially degraded low-resource variants**.
>
> That said, we agree that including a subset of representative SOTA models, specifically those with publicly available and reproducible training code under the 30k split would further strengthen the claims, **but the existing empirical evidence already highlights FEFA's strong data efficiency**. We will incorporate such experiments in the camera-ready version to the extent allowed by computational resources.

---

> ### Author Response · Authors · 2025-12-02
> **Response to Reviewer fkaR (Page 3/4)**
>
> >**W4:** Section 4.3 might be edited for clarity, as it is not immediately clear as to what ablations (a), (b) and (c) represent, or that FEFA is essentially equivalent to MAIRA-2 + (a), (b), (c) and also test-time feedback; since the ablations appear additive, combinations such as error-feedback augmentation without curriculum learning also do not appear to be explored
>
> **A for W4:**
> We thank you for pointing out the ambiguity in our ablation presentation. For clarity, we explicitly define: **(a) refers to adding the Multi-expert Image Fusion module on MAIRA-2; (b) refers to adding Curriculum Learning on (a); (c) refers to adding Error-Aware SFT on (b)**. In addition, FEFA is basically equivalent to MAIRA-2 equipped with the three components above (i.e., (c)) and also the test-time feedback. To improve readability, we have revised the introduction of Section 4.3 (Ablation Study) accordingly.
>
> To address the concern that combinations like "error-feedback without curriculum learning" were unexplored, we performed additional comprehensive ablations on the 5k split (see Table R2).
>
> The results show that **removing any individual component (including curriculum learning) leads to a significant drop in CX-14 performance**, while all ablated configurations still surpass the baseline MAIRA-2 across all metrics. This indicates that **each of our proposed components is both independent and effective**. For example, the curriculum learning is not just an additive bonus but a necessary stabilizer for the model to effectively utilize the error-feedback signal.
>
> These extended experimental results have been added to **Appendix A.3** in the revised manuscript.
>
> **Table R2:** Thorough ablation studies on MIMIC-CXR, which isolate each component we proposed or utilized from the final FEFA. "EFA", "CL", "MEF" are short for error-feedback augmentation, curriculum learning, and multi-expert image fusion, respectively. All models are trained with 5k samples. "FEFA w/o CL & EFA" and "FEFA w/o EFA" have also been shown in Table 4 (i.e., variants (a) and (b)).
>
> |Model|Multi-expert Image Fusion|Curriculum Learning|Error-feedback Augmentation|B-4|MTR|R-L|CX-14 (u$\to$p) Mac-F1|CX-14 (u$\to$p) Mic-F1|CX-14 (u$\to$n) Mac-F1|CX-14 (u$\to$n) Mic-F1|
> |-|-|-|-|-|-|-|-|-|-|-|
> |**FEFA**|✓|✓|✓|5.64|**27.34**|20.93|**35.20**|**47.59**|**30.36**|**47.27**|
> ||||||||||||
> |**FEFA** w/o EFA|✓|✓||4.31|26.21|19.84|31.99|44.62|27.4|44.43|
> |**FEFA** w/o CL|✓||✓|**5.79**|26.44|**21.26**|30.54|40.15|26.13|38.77|
> |**FEFA** w/o MEF||✓|✓|5.34|26.56|20.49|28.28|39.46|22.88|38.00|
> ||||||||||||
> |**FEFA** w/o CL & EFA|✓|||4.42|25.32|19.53|28.27|40.04|23.73|39.89|
> |**FEFA** w/o MEF & EFA||✓||5.43|25.70|20.72|27.44|38.47|22.78|37.35|
> |**FEFA** w/o MEF & CL|||✓|4.84|26.35|20.06|26.64|37.49|22.70|36.72|
> ||||||||||||
> |MAIRA-2||||4.67|23.44|20.28|25.96|36.99|22.27|35.98|
>
> ---
>
> >**Q1:** In Section 3.3, for multi-expert image fusion, it is stated that fused image features are output, if only current images (CF or CF+CL) are available. However, it is then stated that the output is a fixed-length representation. It might be clarified as to whether the Self-Att(.) and Cross-Att(.) dimensionalities are designed to be equal in size, and how this output size was determined.
>
> **A for Q1:**
> We thank you for requesting clarification on this architectural detail.
>
> **Both the Self-Att(.) and Cross-Att(.) modules operate with the same hidden dimension $d$ = 768, matching the channel dimension of the visual encoder outputs (e.g., ViT-Base)**.
>
> **The reason the fused representation is always fixed-length is due to the query-driven design of the attention mechanism.** In standard multi-head attention Att(Q,K,V), the output sequence length and dimensionality are **determined solely by the shape of the Query** (Q).
>
> In our implementation, **the Query for every expert is always projected from the Current Frontal (CF) features of shape $l\times d$ and the auxiliary images (CL or PF) participate only as Keys and Values**.
>
> Consequently, regardless of whether the inputs are single-view (using self-attention where K,V=CF) or multi-view (using cross-attention where K,V=CL), the output feature map strictly preserves the spatial dimension ($l$) and channel dimension ($d$) of the primary CF image. This ensures a consistent input format for the downstream projector and LLM.
>
> To clarify this point, we have specified the shape for the current output features ($l\times d$) in Section 3.3.

---

> ### Author Response · Authors · 2025-12-02
> **Response to Reviewer fkaR (Page 4/4)**
>
> > **Q2:** Related to the above, in Section 4.4, it is then stated that the fused representation contains no more raw information than concatenation, which appears to imply that the size of the fused representation is equal to the maximum size of concatenated views (three views). However, it also appears that the objective should be to maximize FEFA performance (on the NLG and CX-14 metrics), and not to constrain input dimensionality. If so, it could be discussed as to whether a larger fused representation might be appropriate.
>
> **A for Q2:**
> We appreciate your question and clarify that there is a **misunderstanding** regarding the relationship between concatenation size and our fused representation.
>
> Firstly, the fused output in FEFA **does not match the maximum size of concatenated multi-view features**. Instead, FEFA performs substantial dimensionality compression. As the differences are:
>  - Length of concatenation **increases linearly with the number of available views**, i.e., **Length = L_CF + L_CL + L_PF ≈ L to 3L**.
>  - Length of FEFA fusion is **fixed and equal to the primary view length: Length = L_CF = L**.
>
> Therefore, our fused representation **is less than or equal to the size of the concatenated input. This is what we meant by "containing no more raw information"**, the total token space is significantly compressed.
>
> Moreover, you raises an insightful question: Should the fused representation be larger than the primary-view length if the goal is to maximize downstream performance?
>
> In this work, **we chose to align the output dimension strictly with the primary visual encoder's output. This avoids introducing an arbitrary hyperparameter (i.e., the target sequence length) that would require manual tuning or heuristic search**. However, we agree with you that strictly constraining the output to 1L might not be the theoretical upper bound of performance. There could be a "sweet spot" (e.g., 1.5L) where the model balances compactness and capacity.
>
> Exploring expanded or adaptive fused lengths is an interesting direction for future work, and we will note this possibility in the camera-ready version.
>
> ---
>
> > **Q3:** Given the observation on lateral views, it might be clarified as to whether these lateral views are used for clinical diagnosis in current medical practice, and if so, what conditions rely more on lateral views for diagnosis.
>
> **A for Q3:**
> Yes, **lateral views are extensively used in routine medical practice** and are considered part of the "standard of care" for chest radiography. **They are crucial for visualizing areas that are obscured in frontal views** (the so-called "blind spots"), specifically:
>
>  - Retrocardiac Space: Lesions or pneumonia located behind the heart (left lower lobe) are often invisible on frontal views but clear on lateral views.
>  - Posterior Costophrenic Angles: Small amounts of Pleural Effusion are much easier to detect in the deep posterior sulcus seen on lateral views.
>  - Mediastinal & Hilar Masses: Lateral views help localize masses in the anterior/middle/posterior mediastinum.
>
> **We have expanded Section 2 (Related Works) to explicitly clarify why fusing lateral information is medically necessary**. Due to length considerations, only the most important clinical application (identifying hilum abnormalities) is added.
>
> **Reference**
>
> [1] Feigin, David S. "Lateral chest radiograph: a systematic approach." *Academic radiology* 17.12 (2010): 1560-1566.

---

### Official Review · Reviewer_TrBt · 2025-11-01

**Soundness:** 3
**Presentation:** 3
**Contribution:** 2
**Rating:** 4
**Confidence:** 4

**Summary:**

The paper proposes a fusion module using attention and gating mechanisms which shows superior performance over the simpler concatenation scheme used in the state-of-the-art model (MAIRA-2). Additionally, an error-feedback augmentation training strategy is introduced for more robustness.

The approach and specifically the fusion module could be a technical contribution. I'm willing to increase my score if the generalization either on other models or at full training split is convincing.

**Strengths:**

- The writing is easy to follow.
- The method outperforms MAIRA-2 at low resource training splits.
- Comprehensive ablation study.

**Weaknesses:**

- While the training strategy is promising in low resource training splits, the generalization at full training split is unknown.

**Questions:**

- Would the fusion scheme work with multiple prior images? For example, would the approach be able to use prior lateral image if available?
- Would this approach work with other models? What architectures would be applicable?
- Would a reinforcement fine-tuning (RFT) better for error awareness? It seems more prevalent as a method to increase robustness.
- How is the error-feedback augmentation generated? Is it by another specific LLM?

---

> ### Author Response · Authors · 2025-12-02
> **Response to Reviewer TrBt (Page 1/4)**
>
> Thank you for reviewing our paper and for your valuable feedback. Below, we address your concerns point by point and we've revised our paper according to your suggestions.
>
> >**W1:** While the training strategy is promising in low resource training splits, the generalization at full training split is unknown.
>
> **A for W1:**
> **Our study primarily targets the low-resource regime, which reflects the practical constraints of medical applications**. Nevertheless, we agree that examining scalability is important.
>
> **To evaluate generalization under larger training splits, we conducted an additional experiment by doubling the training data from 30k to 60k.** As shown in Table R1, the results exhibit a clear positive scaling trend:
>
>  - With 30k training samples, our model already **achieves 94% of the current SOTA** performance (trained with 400k+ data).
>  - With 60k training samples, the model performance **further improves to 96% on CX-14 and 98% on CX-5. Moreover, both Macro-F1 and Micro-F1 reach similarly high levels (mostly exceeding 95%), and the NLG metrics also show substantial gains**. These results indicate that the model continues to improve as additional data becomes available.
>
> These results show no evidence of saturation and demonstrate that the method continues to benefit from additional supervision.
>
> While we did not train on the full dataset due to the substantial computational cost and our focus on low-resource efficiency, **the trajectory from 30k → 60k suggests that the method would remain competitive, or even surpass existing approaches, at full scale**. Together, these findings confirm that our approach is both scalable and highly data-efficient.
>
> We have revised the paper and put these details in **Appendix A.4.3**.
>
> **Table R1**: Performance of FEFA on MIMIC-CXR trained with 60k samples. CX-5 represents the classification performance for five specific diseases (Atelectasis, Cardiomegaly, Consolidation, Edema, and Pleural Effusion) within CX-14. "%" indicates the ratio relative to the performance of official MAIRA-2. Values above 95% are highlighted in bold.
>
> |Training Scale|Model|B-4|MTR|R-L|CX-14 (u$\to$p) Mac-F1|CX-14 (u$\to$p) Mic-F1|CX-14 (u$\to$n) Mac-F1|CX-14 (u$\to$n) Mic-F1|CX-5 (u$\to$p) Mac-F1|CX-5 (u$\to$p) Mic-F1|CX-5 (u$\to$n) Mac-F1|CX-5 (u$\to$n) Mic-F1|
> |-|-|-|-|-|-|-|-|-|-|-|-|-|
> |30k|**FEFA**|6.84|30.66|22.69|38.93|52.24|34.51|53.18|47.98|54.83|42.96|52.72|
> ||%|49.2%|87.8%|70.8%|90.4%|93.8%|86.6%|94.8%|93.5%|93.3%|91.5%|93.5%|
> |60k|**FEFA**|10.13|31.69|28.02|41.71|53.88|37.25|53.55|50.62|57.31|45.43|54.15|
> ||%|72.9%|90.7%|87.4%|**96.8%**|**96.7%**|93.5%|**95.5%**|**98.6%**|**97.5%**|**96.8%**|**96.1%**|
> |400k+ (Official)|MAIRA-2|13.89|34.93|32.07|43.07|55.70|39.86|56.10|51.34|58.77|46.93|56.37|
> ---

---

> ### Author Response · Authors · 2025-12-02
> **Response to Reviewer TrBt (Page 2/4)**
>
> > **Q1:** Would the fusion scheme work with multiple prior images? For example, would the approach be able to use prior lateral image if available?
>
> **A for Q1:**
> Our main experiments intentionally used only Current Frontal (CF), Current Lateral (CL), and Prior Frontal (PF) to ensure a strictly fair comparison with MAIRA-2 and other SOTA baselines. **Introducing additional views such as Prior Lateral (PL) during training would have provided our method with an unfair information advantage**. Nevertheless, our proposed Multi-Expert Fusion architecture is **fundamentally modular and extensible**, as the experts learn view-specific and time-specific fusion behaviors rather than fixed positional mappings:
>
>  - Spatial Expert (CF-CL/Expert 2) captures cross-view interactions between orthogonal perspectives (Frontal-Lateral). This mechanism naturally generalizes to fusing PF-PL, which share the same cross-view structure.
>  - Temporal Expert (CF-PF/Expert 3) models longitudinal discrepancies between current and historical images. This logic is agnostic to the specific prior view, and can be applied to any available historical image (e.g., PL when PF is missing).
>
> **To validate this extensibility without any retraining, we conducted zero-shot inference experiments using the frozen model on test cases containing PL**, results are shown in Table R2:
>
>  - **Scenario A: PF & PL available**. We reused the Spatial Expert to fuse PF and PL into an Enhanced Prior Feature, which was subsequently processed by the temporal module. The final performance shows a slight degradation, which is expected because the model was not trained to handle this specific input configuration.
>  - **Scenario B: PL only, PF missing**. We applied the Temporal Expert directly to CL and PL, and then fed the resulting features into the Spatial Expert. **The model exceeded the CF+CL baseline, confirming that the temporal reasoning mechanism generalizes to lateral priors even without explicit training supervision for this configuration**.
>
> These zero-shot results confirm that  **our fusion scheme is not limited to specific inputs**. The expert modules can be reused to integrate multiple prior images, including prior lateral views. We expect further performance improvements if such modalities are included during training.
>
> We have revised the paper and put these details in **Appendix A.4.4**.
>
> **Tabel R2:** CX-14 comparisons when inputting prior lateral images (PL) to FEFA (trained with 30k samples) during inference on MIMIC-CXR. "[+CL]" means current lateral images (CL) are optional.
>
> |Input Type|CX-14 (u$\to$n) Mac-F1|CX-14 (u$\to$n) Mic-F1|Input Type|CX-14 (u$\to$n) Mac-F1|CX-14 (u$\to$n) Mic-F1|
> |-|-|-|-|-|-|
> |CF[+CL]+PF (w/ unused PL)|**32.48**|**48.50**|CF+CL (w/ unused PL)|24.78|50.67|
> |CF[+CL]+PF+PL|32.35|47.41|CF+CL+PL|**28.30**|**56.41**|

---

> ### Author Response · Authors · 2025-12-02
> **Response to Reviewer TrBt (Page 3/4)**
>
> > **Q2:** Would this approach work with other models? What architectures would be applicable?
>
> **A for Q2:**
> **Our approach is designed to be highly generic and compatible with modern Multimodal LLM (MLLM) architectures** that follows the widely adopted Vision Encoder → Feature Projector → LLM pipeline. Since,
>
> - Our fusion module operates on the visual feature space before the tokens are fed into the LLM, it functions as a plug-and-play refinement layer that is **independent** of the decoder architecture, tokenizer, or training objective.
> - The Error-Feedback Strategy is a data-driven instruction tuning method, it is **universally applicable** to any autoregressive LLM capable of supervised fine-tuning.
>
> To empirically validate this generality, **we extended our evaluation to Microsoft's Phi-3-mini-4k-instruct and Phi-4-mini-instruct**. These models represent a different architectural family and a smaller parameter scale (3.8B) compared to our main backbone (Vicuna-7B-v1.5).
>
> As shown in Table R3, our method was compared with the standard Direct Concatenation baseline on both new backbones. **The results remained consistent with our results in main paper**.
>
> - Our fusion module outperformed the concatenation baseline on both Phi-3 and Phi-4 across all metrics. This **indicates that the underlying issue we address (visual redundancy in multi-view medical imaging) is intrinsic to the data rather than tied to a specific decoder architecture**.
>
> - We noted that the absolute scores on the Phi models (3.8B) were slightly lower than on Vicuna (7B). This is **an expected outcome due to the reduced model capacity (parameter count)**.
>
> These experiments demonstrate that our contribution is not tailored to a specific model size or family (e.g., LLaMA-based Vicuna). Instead, it serves as a generalized optimization framework applicable to a wide range of MLLM architectures.
>
> We have revised the paper and put these details in **Appendix A.4.2**.
>
> **Tabel R3:** Performance comparisons with SOTA MAIRA-2 on different LLM backbones on MIMIC-CXR (all trained with 30K samples).
>
> |Model|B-4|MTR|R-L|CX-14 (u$\to$p) Mac-F1|CX-14 (u$\to$p) Mic-F1|CX-14 (u$\to$n) Mac-F1|CX-14 (u$\to$n) Mic-F1|
> |-|-|-|-|-|-|-|-|
> |MAIRA-2 (w/ Vicuna-7B v1.5)|6.36|27.60|22.55|33.50|46.72|28.86|46.80|
> |**FEFA** (w/ Vicuna-7B v1.5)|**6.84**|**30.66**|**22.69**|**38.93**|**52.24**|**34.51**|**53.18**|
> |MAIRA-2 w/ Phi-3-mini [1]|4.11|24.88|17.19|16.80|28.86|12.48|25.45|
> |**FEFA** w/ Phi-3-mini [1]|**4.45**|**27.59**|**17.23**|**30.21**|**42.38**|**26.86**|**42.49**|
> |MAIRA-2 w/ Phi-4-mini [2]|2.14|19.55|12.20|15.10|27.10|12.80|26.54|
> |**FEFA** w/ Phi-4-mini [2]|**5.53**|**31.12**|**19.24**|**35.17**|**47.93**|**31.59**|**48.90**|
>
> **References**
>
> [1] Abdin, Marah et al. "Phi-3 Technical Report: A Highly Capable Language Model Locally on Your Phone." *arXiv preprint arXiv:2404.14219* (2024).
>
> [2] Abouelenin, Abdelrahman, et al. "Phi-4-mini technical report: Compact yet powerful multimodal language models via mixture-of-loras." *arXiv preprint arXiv:2503.01743* (2025).
>
> ---
> > **Q3:** Would a reinforcement fine-tuning (RFT) better for error awareness? It seems more prevalent as a method to increase robustness.
>
> **A for Q3:**
>
> We appreciate your insightful suggestion. We agree that Reinforcement Fine-Tuning (RFT), such as RLHF or RLAIF, which is indeed an increasingly popular strategy for improving robustness and error awareness.
>
> While RFT is indeed promising, it typically **requires substantial resources** for reward-model training, policy optimization, and stability control. **One of the core motivations of our work is to design a method that remains effective under low-resource constraints, where such reinforcement pipelines may be impractical**. Our error-feedback mechanism offers **a lightweight alternative** that introduces error awareness without requiring reward models or iterative rollouts, **making it far more accessible to medical institutions with limited computational budgets**.
>
> Moreover, **a strong and well-structured SFT initialization is widely acknowledged as a prerequisite** for successful reinforcement fine-tuning. **We view our approach not as an alternative to RFT but as a necessary precursor. A model that has already internalized error awareness during SFT provides a more stable and effective starting point for subsequent RFT**. As you suggested, integrating RFT to further refine this robustness is indeed the next step in our research roadmap, and preliminary investigations are already underway.
>
> In summary, while RFT is a valuable avenue for future improvement, our results show that a carefully designed SFT strategy can already yield substantial gains in error awareness with significantly lower computational overhead, and can serve as a strong foundation for future reinforcement fine-tuning.

---

> ### Author Response · Authors · 2025-12-02
> **Response to Reviewer TrBt (Page 4/4)**
>
> > **Q4:** How is the error-feedback augmentation generated? Is it by another specific LLM?
>
> **A for Q4:**
> Thank you for raising this question. We realize the manuscript may not have sufficiently highlighted how the error-feedback augmentation is generated, and we appreciate the opportunity to clarify.
>
> First, **the error feedback is not generated by any external LLM** (such as GPT-4). Instead, **our method adopts a self-supervised correction mechanism, where the model learns from its own prediction errors**. This design keeps the pipeline lightweight and aligned with the limited-resource setting of our work.
>
> **As shown in Figure 3 and described in Section 3.5 of the main paper, the feedback is constructed through the following steps**:
>
>  - Step 1 (Inference): We use the model trained in the second stage (without feedback) to generate predicted reports for the entire training set.
>  - Step 2 (Labeling & Comparison): A standard clinical labeler (CheXbert) extracts the statuses of 14 clinical observations from both the Predicted Reports and the Ground Truth Reports.
>  - Step 3 (Structured Feedback Construction): We identify discrepancies such as missed findings or hallucinated findings. These errors are then converted into structured textual "Error Feedback" prompts, which augment the training samples for the third-stage supervised fine-tuning.
>
> This closed-loop mechanism ensures that the feedback directly reflects the model's own weaknesses rather than generic or externally generated annotations, thereby enabling the model to effectively "learn from its own mistakes."

---

### Official Review · Reviewer_g6KK · 2025-11-01

**Soundness:** 3
**Presentation:** 3
**Contribution:** 3
**Rating:** 6
**Confidence:** 3

**Summary:**

This paper proposes FEFA, a method designed for low-resource radiology report generation (RRG). The framework integrates:

- A multi-expert image fusion module combining current, lateral, and prior images with attention and gating mechanisms.

- An error-feedback augmentation (EFA) strategy inspired by large language model (LLM) self-correction, introducing feedback from earlier training stages as additional textual prompts.

Experiments on MIMIC-CXR show that FEFA achieves ~94% of the full-data MAIRA-2 clinical efficacy score while using only 8% of supervised data. The paper claims improvements in data efficiency and model adaptability.

**Strengths:**

1.Timely and relevant topic — Tackling low-resource RRG aligns well with the current push toward efficient and clinically reliable medical AI.

2.Novel integration of LLM feedback — Applying error-feedback augmentation for radiology generation is an interesting direction.

3.Reasonable empirical results — The reported gains on MIMIC-CXR under low-resource settings are consistent and moderately significant.

4.Good presentation and writing quality — Figures and tables are clear, methodology is described in a readable manner.

**Weaknesses:**

1.Lack of technical depth and originality (Major):The multi-expert fusion design appears to be a modest extension of MAIRA-2’s concatenation scheme. The attention/gating mechanism lacks architectural novelty and seems incremental. The error-feedback augmentation mainly reuses textual prompts derived from model outputs — similar ideas have been extensively explored in Self-Refine (Madaan et al., 2023), Reflexion (Shinn et al., 2023), and LEMA (An et al., 2023). There is no theoretical or algorithmic innovation specific to the medical domain. The method is essentially a composition of known modules without deep insight into why these components interact effectively.

2.Weak experimental validation (Major): Experiments are only conducted on a single dataset (MIMIC-CXR), which severely limits generalizability. No external dataset (e.g., IU-XRay, CheXpert-Plus) or real-world validation is provided. The claimed “8% data efficiency” lacks ablation under different data sampling strategies — it’s unclear whether the gain persists under non-uniform splits.

3.Unclear evaluation methodology: No human expert evaluation or radiologist assessment is provided to validate clinical soundness. NLG metrics (BLEU, ROUGE, METEOR) remain low compared to state-of-the-art models.

4.Weak connection between error feedback and performance gain: The claimed “self-correction” ability is not convincingly demonstrated. No quantitative analysis shows that the model reduces specific error types after feedback integration.

5.Limited theoretical grounding (Minor): The Uniform Information Density (UID) analysis is interesting but speculative; it lacks rigorous proof or empirical correlation with model performance.

**Questions:**

See weaknesses

---

> ### Author Response · Authors · 2025-12-02
> **Response to Reviewer g6KK (Page 1/7)**
>
> Thank you for your valuable feedback to help us improve our paper. We have revised our paper based on your feedback. We detail our response below and please kindly let us know if our response addresses your concerns.
>
> >**W1-1:** Lack of technical depth and originality (Major): The multi-expert fusion design appears to be a modest extension of MAIRA-2's concatenation scheme. The attention/gating mechanism lacks architectural novelty and seems incremental.
>
> **A for W1-1:**
> We respectfully clarify that our multi-expert fusion module is not a modest extension of MAIRA-2's concatenation scheme. We articulate our technical depth and novelty from **three** distinct perspectives:
>
> - Fundamental paradigm shift (beyond concatenation): Our design represents a fundamental shift from **passive data splicing** (direct concatenation) to **active retrieval of complementary information**.
>   - Unlike concatenation (e.g., MAIRA-2) which treats all views as undifferentiated inputs, our architecture explicitly **encodes the "radiological reasoning prior"** — clinicians use current frontal images as an anchor to **query spatial depth from lateral views and temporal changes from prior views**.
>   - We implemented this by treating the current frontal view as the "Query". This transforms the fusion process from linear token stacking (concatenation) into targeted extraction of signals, fundamentally altering how visual information is processed in RRG tasks.
> - Structural novelty & validation: For your expressed concern that the attention/gating mechanism seems incremental. We argue that the **novelty lies not in the atomic operations (attention/gating), but in the topology of expert composition (spatial vs. temporal disentanglement)**.
>   - To validate the architectural contribution (not just "adding attention/gating"), we conducted a controlled study on the 15k split. We replaced our multi-expert module with **a generic simple cross-attention module**, which treats all auxiliary images as a single flat context (similar to standard non-expert approaches).
>   - As shown in Table R1, the generic design performs significantly **worse** than our multi-expert design (but still better than MAIRA-2). This confirms that generic mechanisms fail to capture the nuanced clinical distinctions between "spatial depth" and "temporal progression." Our specific architectural separation is technically non-trivial and essential for performance.
> - Problem-driven Innovation (i.e., solving existed flaws): Our work is not incremental because it specifically targets and resolves critical bottlenecks identified in most prior works like MAIRA-2.
>   - **We are the first to identify and analyze MAIRA-2's inefficiency in leveraging lateral information.** As shown in Tables 1 and 5 in the main paper, **compared to MAIRA-2, our multi-expert fusion module can significantly better utilize lateral images to complement frontal images**.
>   - Most prior works like MAIRA-2 suffers from excessive sequence length and visual noise. Our method solves this by maintaining a fixed-length, high-density, uniform output.
>
> Therefore, based on the above, we believe that our multi-expert fusion module has already provided a substantial and non-incremental advancement.
>
> **Table R1:** Performance comparisons between multi-expert fusion (our attention/gating mechanism) and simple cross-attention.
>
> |Model|B-4|MTR|R-L|CX-14 (u$\to$p) Mac-F1|CX-14 (u$\to$p) Mic-F1|CX-14 (u$\to$n) Mac-F1|CX-14 (u$\to$n) Mic-F1|
> |-|-|-|-|-|-|-|-|
> |MAIRA-2|**5.02**|24.61|**21.19**|27.76|39.31|24.18|39.15|
> |Simple Cross-attention|4.41|25.07|19.11|29.54|40.89|25.96|41.14|
> |Multi-expert Fusion|4.56|**25.99**|19.05|**32.45**|**44.86**|**26.95**|**44.60**|

---

> ### Author Response · Authors · 2025-12-02
> **Response to Reviewer g6KK (Page 2/7)**
>
> >**W1-2:** The error-feedback augmentation mainly reuses textual prompts derived from model outputs - similar ideas have been extensively explored in Self-Refine (Madaan et al., 2023), Reflexion (Shinn et al., 2023), and LEMA (An et al., 2023).
>
> **A for W1-2:**
> Thank you for connecting our work to these prominent studies. While we share the broad motivation of utilizing error signals to improve generation, **our approach diverges fundamentally from Self-Refine** (Madaan et al., 2023), **Reflexion** (Shinn et al., 2023), **and LEMA** (An et al., 2023) **in terms of implementation phase, efficiency, and dependency**.
>
> The works mentioned (Self-Refine, Reflexion) function primarily **at inference time**, requiring the model to generate, critique, and regenerate iteratively for every single test sample. **This "multi-step reasoning" significantly increases inference latency and computational cost.**
>
> In contrast, our method treats error feedback as additional input **during training** via instruction tuning. By exposing the model to its own historical errors and the corresponding corrections during training, we internalize the self-correction capability into the model's weights. **Consequently, at test time, our model performs standard single-pass inference, achieving high accuracy without the overhead of iterative refinement steps.** This shift from inference-time correction to training-time internalization is a core methodological distinction.
>
> Moreover, many existing feedback frameworks (like LEMA or Reflexion) often **rely on a stronger "Teacher" model (e.g., GPT-4) or external verifiers to provide high-quality critique. Our framework is entirely endogenous**. The feedback is derived from the model's own previous stage predictions compared against ground truth labels. This creates a self-contained optimization loop that does not require access to superior external models or knowledge bases, **making it highly suitable for privacy-sensitive medical domains where external APIs may be restricted.**
>
> We have briefly discussed these distinctions in Section 2 (Related Works) and Section 3.5 (Error-feedback Augmentation) due to the length limit. If allowed, we will further highlight the "training-time self-correction internalization" aspect in the camera-ready version.

---

> ### Author Response · Authors · 2025-12-02
> **Response to Reviewer g6KK (Page 3/7)**
>
> >**W1-3:** There is no theoretical or algorithmic innovation specific to the medical domain. The method is essentially a composition of known modules without deep insight into why these components interact effectively.
>
> **A for W1-3:**
> We respectfully disagree with the assessment that the method lacks domain-specific innovation or deep insight into component interaction.
>
> While the individual building blocks (e.g., curriculum learning, SFT) are established, our **contribution in how these components are strategically composed to address the algorithmic challenges unique to medical report generation**.
>
> - We **model the diagnostic reasoning process of real clinicians** by **explicitly encoding the radiological reasoning prior** — physicians use the current frontal image as the anchor to query **spatial depth** from the lateral view and **temporal progression** from prior studies. We achieved this by utilizing a spatial expert (Expert 2) to capture cross-view interactions and a temporal expert (Expert 3) to model longitudinal discrepancies, respectively.
>
> - Our fusion module is deliberately designed to **decouple medical views**. Through a query-driven expert mechanism, the model **extracts only the complementary clinical signals** (e.g., depth cues from Lateral, progression from Prior) without inflating the sequence length. This design not only preserves high information density but also **enables the effective incorporation of additional modalities** such as prior lateral images, which standard concatenation approaches struggle to handle efficiently. Therefore, **our method provides a scalable and principled pathway** for future multi-view medical imaging research.
>
> Moreover, our system is not a loose collection of independent modules. Instead, its components form a **synergistic, bottom-up optimization pipeline**, each improving the model from a different perspective:
>
> - Multi-expert Fusion provides the initial "clean signal" with high UID. Without this, the downstream LLM would overfit to redundant visual noise.
>
> - Curriculum Learning optimizes the *core training dynamics*, helping the model first align visual and textual features before it can effectively handle higher-level logical corrections.
>
> - Error feedback serves as the final logic correction, addressing systematic errors that visual features alone cannot resolve.
>
> The components are highly interactive in terms of optimization dynamics, **each stage establishes a stronger foundation for the next, and together they form a mutually complementary optimization structure**. As shown in Table R2, **although they can function as stackable plugins, they achieve optimal performance when working as a cohesive system**.
>
> **Table R2:** Thorough ablation studies on MIMIC-CXR, which isolate each component we proposed or utilized from the final FEFA. "EFA", "CL", "MEF" are short for error-feedback augmentation, curriculum learning, and multi-expert image fusion, respectively. All models are trained with 5k samples. "FEFA w/o CL & EFA" and "FEFA w/o EFA" have also been shown in Table 4 (i.e., variants (a) and (b)).
>
> |Model|Multi-expert Image Fusion|Curriculum Learning|Error-feedback Augmentation|B-4|MTR|R-L|CX-14 (u$\to$p) Mac-F1|CX-14 (u$\to$p) Mic-F1|CX-14 (u$\to$n) Mac-F1|CX-14 (u$\to$n) Mic-F1|
> |-|-|-|-|-|-|-|-|-|-|-|
> |**FEFA**|✓|✓|✓|5.64|**27.34**|20.93|**35.20**|**47.59**|**30.36**|**47.27**|
> ||||||||||||
> |**FEFA** w/o EFA|✓|✓||4.31|26.21|19.84|31.99|44.62|27.4|44.43|
> |**FEFA** w/o CL|✓||✓|**5.79**|26.44|**21.26**|30.54|40.15|26.13|38.77|
> |**FEFA** w/o MEF||✓|✓|5.34|26.56|20.49|28.28|39.46|22.88|38.00|
> ||||||||||||
> |**FEFA** w/o CL & EFA|✓|||4.42|25.32|19.53|28.27|40.04|23.73|39.89|
> |**FEFA** w/o MEF & EFA||✓||5.43|25.70|20.72|27.44|38.47|22.78|37.35|
> |**FEFA** w/o MEF & CL|||✓|4.84|26.35|20.06|26.64|37.49|22.70|36.72|
> ||||||||||||
> |MAIRA-2||||4.67|23.44|20.28|25.96|36.99|22.27|35.98|

---

> ### Author Response · Authors · 2025-12-02
> **Response to Reviewer g6KK (Page 4/7)**
>
> >**W2:** Weak experimental validation (Major): Experiments are only conducted on a single dataset (MIMIC-CXR), which severely limits generalizability. No external dataset (e.g., IU X-ray, CheXpert-Plus) or real-world validation is provided. The claimed "8% data efficiency" lacks ablation under different data sampling strategies - it's unclear whether the gain persists under non-uniform splits.
>
> **A for W2:**
> We thank you for pointing out the need for broader validation. We take this concern seriously and **have conducted two additional sets of experiments to rigorously validate both cross-domain robustness and sampling-split stability**.
>
> **To prove that our method is not overfitted to MIMIC-CXR, we evaluated our model on the IU X-ray dataset**, which differs substantially from MIMIC-CXR in imaging protocols, patient demographics, and reporting style. As shown in Table R3, consistent with our main findings, **FEFA again significantly outperformed the direct concatenation baseline on IU X-ray**. Remarkably, even in this cross-domain setting, our method retained approximately 93% of the performance of the heavy SOTA baseline (MAIRA-2) under the low-resource 30k regime, **mirroring the efficiency observed on MIMIC-CXR**. This confirms that our multi-expert fusion learns invariant radiological features rather than dataset-specific biases.
>
> To address the concern that the reported "8% data efficiency" could be due to a favorable uniform split, **we conducted a stress test using a smaller and biased subset**. We reduced the data scale to 15k (approx. 4%) and **applied a non-uniform (imbalanced) sampling strategy that preserves the label distribution ratios of the original dataset** (CF: 8.72%, CF+CL: 26.66%, CF+PF: 32.35%, CF+CL+PF: 32.27%). As shown in Table R4, **the model trained on this imbalanced 15k split achieved performance statistically comparable to the model trained on a balanced 15k split**. Since the model remains robust under this extreme setting (4% data + imbalance), we can reasonably infer that at larger scales (8%/30k or more), the impact of sampling variance would be more negligible. This confirms that the reported efficiency gains stem from the proposed architecture, rather than the data sampling strategy.
>
> We have revised the paper and put these details in **Appendix A.2 and A.4.1**.
>
> **Table R3**: Performance comparisons with SOTA MAIRA-2 on varing low-resource training sets on IU X-ray.
>
> |Training Scale|Model|B-4|MTR|R-L|CX-14 (u$\to$p) Mac-F1|CX-14 (u$\to$p) Mic-F1|CX-14 (u$\to$n) Mac-F1|CX-14 (u$\to$n) Mic-F1|
> |-|-|-|-|-|-|-|-|-|
> |5k|MAIRA-2|6.06|32.33|23.33|9.42|23.88|8.84|25.43|
> ||**FEFA**|**7.62**|**34.64**|**24.92**|**19.83**|**46.20**|**18.83**|**48.72**|
> |15k|MAIRA-2|6.50|32.45|23.93|13.67|35.45|12.48|38.15|
> ||**FEFA**|**7.33**|**33.87**|**25.27**|**22.66**|**46.77**|**20.75**|**49.41**|
> |30k|MAIRA-2|6.74|32.75|24.47|21.27|35.09|18.88|37.49|
> ||**FEFA**|**7.22**|**34.79**|**25.93**|**27.15**|**46.76**|**25.82**|**49.48**|
> |400k+ (Official)|MAIRA-2|8.29|33.67|28.30|31.29|50.44|29.34|52.56|
>
> **Table R4**: Performance comparisons under different sampling strategies of **FEFA** (trained with 15k samples) on MIMIC-CXR.
>
> |Sampling Strategy|B-4|MTR|R-L|CX-14 (u$\to$p) Mac-F1|CX-14 (u$\to$p) Mic-F1|CX-14 (u$\to$n) Mac-F1|CX-14 (u$\to$n) Mic-F1|
> |-|-|-|-|-|-|-|-|
> |uniform (equal for each input type, **default**)|6.54|29.65|22.32|**38.15**|51.08|**33.32**|**51.17**|
> |non-uniform (randomly in the training set)|**7.24**|**30.51**|**23.07**|38.05|**51.25**|31.55|50.12|

---

> ### Author Response · Authors · 2025-12-02
> **Response to Reviewer g6KK (Page 5/7)**
>
> > **W3:** Unclear evaluation methodology: No human expert evaluation or radiologist assessment is provided to validate clinical soundness. NLG metrics (BLEU, ROUGE, METEOR) remain low compared to state-of-the-art models.
>
> **A for W3:**
> We appreciate your emphasis on clinical soundness. We would like to clarify our evaluation standards and the context of our metric scores.
>
> First, **large-scale human evaluation is not feasible for most RRG works in reality**, unfortunately. For example, no baseline we compared with in Table 2 adopted human evaluation except the official MAIRA-2 . Under this circumstance, **we strictly followed community standards established by prior work** (e.g., MAIRA-2 [1], R2Gen [2]) by **prioritizing Clinical Efficacy Metrics (CX-14) over pure NLG metrics**. CheXbert is a rigorous metric trained on expert-labeled data to detect the presence/absence of 14 distinct pathologies. It serves as a validated "AI Radiologist Proxy" to measure clinical soundness and **is widely accepted in the research community as robust proxies** that correlate highly with expert consensus on diagnostic accuracy.
>
> Second, **the comparatively lower NLG scores should be interpreted in light of our low-resource setting**. The "low" scores arise primarily because we trained on only 8% (30k) of the data, whereas the SOTA models use 100% (400k+). **Comparing raw BLEU scores directly against full-data models is inherently skewed**. To demonstrate that our architecture is not the bottleneck, we refer to our new experiment on the 16% (60k) split. **As shown in Table R5, with this moderate data increase, our NLG metrics improved significantly, narrowing the gap with SOTA**.
>
> Furthermore, the results in Table R3 show that **on the IU X-ray dataset, our method not only matched but in MTR surpassed the official MAIRA-2 in NLG metrics**. This confirms that our model's language generation capability is robust and scalable.
>
> Finally, **in medical reporting, Clinical Accuracy (CX-14) where we excel is often considered more critical than NLG metrics** which are known to be weakly aligned with radiologist assessments, as distinct phrasings can describe the same correct diagnosis. **As shown in Section 4.6 and the Appendix A.7, we have provided case studies demonstrating that FEFA yields fluent and clinically coherent descriptions**.
>
> **Table R5**: NLG metrics' improvements of FEFA on MIMIC-CXR trained with 60k samples compared to 30k samples. "%" indicates the ratio relative to the performance of official MAIRA-2.
>
> |Training Scale|Model|B-4|MTR|R-L|
> |-|-|-|-|-|
> |30k|**FEFA**|6.84|30.66|22.69|
> ||%|49.2%|87.8%|70.8%|
> |60k|**FEFA**|10.13|31.69|28.02|
> ||%|72.9%|90.7%|87.4%|
> |400k+ (Official)|MAIRA-2|13.89|34.93|32.07|
>
> **References**
>
> [1] Bannur, Shruthi, et al. "Maira-2: Grounded radiology report generation." *arXiv preprint arXiv:2406.04449* (2024).
>
> [2] Chen, Zhihong, et al. "Generating radiology reports via memory-driven transformer." *EMNLP 2020*, pp. 1439–1449 (2020).

---

> ### Author Response · Authors · 2025-12-02
> **Response to Reviewer g6KK (Page 6/7)**
>
> >**W4:** Weak connection between error feedback and performance gain: The claimed "self-correction" ability is not convincingly demonstrated. No quantitative analysis shows that the model reduces specific error types after feedback integration.
>
> **A for W4:**
> We appreciate your request for a granular breakdown of the error-feedback mechanism. **To convincingly demonstrate that the model reduces specific error types rather than just achieving random gains, we performed a controlled sensitivity analysis with noise injection** (results are shown in Table R6).
>
> Specifically, **we created three tiers of feedback quality by varying the mismatch frequency between second-stage predictions and ground truth**:
>  - Tier 1 (high-quality): Prompts containing the top-4 high-frequency, systematic error types ("true pain points").
>  - Tier 2 (mild noise): Prompts containing 4 medium-frequency, partially relevant error types.
>  - Tier 3 (severe noise): Prompts containing the bottom-4 low-frequency error types, simulating noisy, inconsistent, or hallucinated feedback.
>
> All experiments were conducted using the 30k model, evaluating the 14 CheXbert labels (treating uncertain labels as negative). The results (Table R5) reveal three critical findings that validate the "self-correction" capability:
>
>  - **Adding error feedback at any frequency tier yielded a higher average performance** compared to the baseline without feedback.
>  - For all tiers, **the performance gain for labels explicitly mentioned in the feedback prompt was consistently higher** than for those unmentioned. This quantitatively proves that the **model is actively attending to the feedback and reducing the specific error types it was instructed to fix**.
>  - We observed a distinct behavior regarding unmentioned labels. Under Tier 1, the correction of high-frequency systematic errors led to a positive spillover effect, **improving unmentioned labels as well**. But under noise (Tier2 and Tier3), **while the model successfully improved the specific low-frequency errors it was told to focus on, this came at the cost of degrading the unmentioned labels**.
>
> These results collectively reinforce two conclusions:  (1) **The model indeed possesses a strong ability to perform targeted self-correction**; (2) **Our proposed strategy (Tier 1) is essential** because it aligns the feedback with the model's actual systematic deficits, achieving a holistic improvement without the negative side effects seen in noisy settings.
>
> We have revised the paper and put these details in our **new Section 4.4**.
>
> **Table R6**: Performance gains of FEFA (trained with 30k samples) across each clinical observation in CX-14 when using errors of different frequencies as feedback during inference. "Obs." is short for observations and positive $\Delta$Acc. are highlighted in bold.
>
> |14 Clinical Observations Labeled by CheXbert|FEFA w/o Error-aware Prediction|FEFA (w/ Most Frequent Errors)||FEFA w/ Moderate Frequent Errors||FEFA w/ Least Frequent Errors||
> |-|-|-|-|-|-|-|-|
> ||Accuracy|$\Delta$Acc.|Obs. in Feedback|$\Delta$Acc.|Obs. in Feedback|$\Delta$Acc.|Obs. in Feedback|
> |Enlarged Cardiomediastinum|87.85|**0.16%**|✓|-1.99%|✓|-1.54%||
> |Cardiomegaly|70.58|-2.03%|✓|-1.95%|✓|-2.28%||
> |Lung Opacity|62.82|**1.14%**|✓|**0.24%**||**0.20%**|✓|
> |Lung Lesion|92.32|**0.93%**||**1.06%**|✓|**0.53%**|✓|
> |Edema|79.20|**1.30%**|✓|**1.18%**|✓|**0.08%**|✓|
> |Consolidation|92.32|**1.63%**||-0.08%|✓|**0.16%**|✓|
> |Pneumonia|93.25|**1.46%**|✓|**1.38%**|✓|**0.61%**|✓|
> |Atelectasis|69.08|**1.22%**|✓|-1.95%||**1.02%**||
> |Pneumothorax|96.46|**1.46%**||**0.81%**|✓|**1.02%**|✓|
> |Pleural Effusion|76.68|**1.58%**|✓|**0.49%**|✓|-0.45%||
> |Pleural Other|95.73|**0.65%**||**0.37%**|✓|-0.20%|✓|
> |Fracture|94.80|-0.41%||-0.73%|✓|-1.10%|✓|
> |Support Devices|84.19|-0.08%||-1.26%|✓|-0.65%||
> |No Finding|88.58|**1.02%**|✓|**3.01%**|✓|**2.93%**||
> |**Avg. of All Obs.**|*84.56*|*0.72*%||*0.04%*||*0.02%*||
> |**Avg. of Obs. in Feedback**|-|*0.73%*||*0.19%*||*0.16%*||
> |**Avg. of Obs. Not in Feedback**|-|*0.70%*||*-0.85%*||*-0.16%*||

---

> ### Author Response · Authors · 2025-12-02
> **Response to Reviewer g6KK (Page 7/7)**
>
> >**W5:** Limited theoretical grounding (Minor): The Uniform Information Density (UID) analysis is interesting but speculative; it lacks rigorous proof or empirical correlation with model performance.
>
> **A for W5:**
> We appreciate your comment, but we need to clarify some facts.
>
> UID theory is a cognitive hypothesis widely used in NLP. It suggests that more uniform information density reduces cognitive load for human readers, thereby improving comprehension, and provides a quantitative lens to analyze cognitive efficiency. In this work (Section 4.5), **we argue that the performance improvement brought by our fusion module stems from the fact that its output features are easier for the LLM to process** (it reduces the cognitive load imposed on the LLM). **UID offers a principled framework to interpret this improvement.** We adapt UID to multimodal medical report generation by treating the fusion module as the information sender and the downstream LLM as the information receiver. This framework allows us to interpret why our fusion strategy is effective as: **more uniform image features’ information density → lower LLM cognitive load → higher downstream performance.** And **this is the empirical correlation between UID and model performance**.
>
> While UID theory itself is widely used in practice, it has not been rigorously proven, therefore, we cannot provide a rigorous proof. Instead, we conducted **multiple sets of quantitative experiments** to demonstrate that the premises in this conclusion hold true, thereby achieving **our initial goal: to provide interpretability for the performance improvement** of the model caused by the fusion module.
>
> - In **Section 4.5** of the main paper, we **directly quantify UID using the standard deviation of entropy**. Across all test samples, our fusion module consistently yields lower entropy variance, meaning a more uniform information stream than direct concatenation.
> - In **Appendix A.5**, we quantitatively analyze LLM attention behaviors, which **indirectly demonstrate that the uniformity of the incoming image features**.
>   - Experiments (Figure 6 and Figure 7) show that our top-5 attention mass is lower for all test samples. This demonstrates that our LLM does not require excessive information filtering when processing image features, allowing for a **smoother distribution of attention** across semantically relevant features. In contrast, in direct concatenation, LLM needs to extract effective information from sequences containing a large amount of redundancy and noise, increasing the model's burden.
>   - We also added attention heatmaps (Figure 8) to show the degree of attention given to the input sequence, which more intuitively illustrates that our LLM exhibits **distributed rather than spiky attention**, avoiding bottlenecks and again reflecting more uniform input features.
>
> Since the dataset and task are identical, **the improvement in feature uniformity is directly linked to the performance gain**. This has provided the "empirical correlation" requested.

---

### Author Response · Authors · 2025-12-02
**Summary of Paper Revision**

We sincerely appreciate all reviewers for their insightful and constructive feedback. Based on these comments, we have improved the paper (new pdf uploaded) and highlighted the main changes with blue text. Importantly, **we did not modify any experimental conclusions or numerical results**. This revision focuses on clarifying misunderstandings and adding supplementary analyses requested by the reviewers. Below, we summarize all newly added experiments:

- In Section 4.4, we added accuracy comparisons of our model across each clinical observation in CX-14 under three feedback strategies.
- In Appendix A.2, we added performance comparisons on the IU X-ray dataset.
- In Appendix A.3, we added a more comprehensive ablation analysis on the components of our model.
- In Appendix A.4, we conducted additional experiments to demonstrate the robustness and scalability of our model.
  - In Appendix A.4.1, we added our performance under a non-uniform sampling strategy.
  - In Appendix A.4.2, we added performance comparisons of using alternative LLM backbones.
  - In Appendix A.4.3, we added our performance with more training samples (i.e., 60k).
  - In Appendix A.4.4, we added our performance when integrating prior lateral (PL) views during inference.
- In Appendix A.5, we added an additional analysis based on “surprisal” in UID theory and presented corresponding attention heatmaps in Figure 8.

---

### Meta-Review · Area_Chair_yHrr · 2025-12-31

**Summary:**

The paper suffers from major shortcomings that undermine its claims. Most critically, the experimental comparison with prior work is unfair and inconsistent in evaluation metrics: in Table 2, FEFA is reported using micro‑average scores, whereas several baselines (e.g., ICON) use macro‑average scores, which are not directly comparable. This metric mismatch biases the perceived improvement. In addition, the chosen baseline (MAIRA‑2) in the 30K setting is weak, with Mac‑F1 only 33.50, while other existing methods such as PromptMRG achieved 47+ Mac‑F1 on similar tasks and data scales. The reported gain from 33.50 to 39.45 therefore does not convincingly indicate superiority over the real state‑of‑the‑art. The scope of evaluation is narrow, lacking tests on stronger baselines under identical data and metric setups, and missing broader datasets such as CheXpert‑Plus. Given the unfair comparison, inconsistent metrics, inflated performance claims, and unresolved methodological concerns, the AC recommends rejecting this paper at this stage.

**Reviewer Concerns:**

In terms of cross-domain generalization, error-feedback mechanism effectiveness and robustness, curriculum learning independent contribution, and architecture and training details clarification, the rebuttal provides relatively clear responses. However, concerns still remain regarding fairness and strong baseline comparison, novelty concerns, and the lack of clinical quality validation.

**Reviewer Scores:**

Given the still unresolved concerns, the reviewers may have no clear motivation to raise their scores.

---

> ### Public Comment · ~YiMing_Ji2 · 2026-03-08
>
> Thank you for your time and effort in handling our submission. We fully respect the review process and value the feedback provided. Our purpose in this comment is solely to clarify a factual point in the meta-review that may have affected the evaluation of our work, to ensure accurate interpretation for current and future submissions.
>
> In the meta-review, our results under the 30K-sample, low-resource setting were compared with PromptMRG, and the performance gap was interpreted as evidence that our method does not reach state-of-the-art under similar data conditions. Upon careful review, we would like to clarify that PromptMRG was trained on the full MIMIC-CXR dataset (~270K studies), whereas our study explicitly focuses on a low-resource regime with only 30K samples. This nearly 9× difference in supervision scale makes the two settings not directly comparable, and using such a comparison as the basis for assessing relative performance may lead to a misleading conclusion.
>
> Additionally, the meta-review cites PromptMRG’s reported F1 of 0.476 from the main text. This metric corresponds to an example-based clinical efficacy F1, rather than the Micro-F1 or Macro-F1 commonly used in radiology report generation evaluation. Comparing this example-based F1 directly with our Macro-F1 could result in a misunderstanding of relative performance. In fact, PromptMRG reports a Macro-F1 of 38.1 in the appendix.
>
> We fully understand that, given the large number of submissions and space constraints, such misunderstandings are understandable. We do not question the review process itself; our sole intent is to clarify that this specific comparative point was based on an incorrect assumption about training data scale and metric type. Clarifying this point is important to ensure that future evaluations of similar methods are based on accurate factual understanding.
>
> We sincerely hope that this clarification can be taken into consideration. Regardless of outcome, we remain grateful for the feedback and will use it to improve the clarity of our work.

---

### Decision · Program_Chairs · 2026-01-26

Reject